# Regional validation of the use of diatoms in ice cores from the Antarctic Peninsula as a Southern Hemisphere Westerly Wind proxy

Dieter R. Tetzner[1, 2], Elizabeth R. Thomas[1], Claire S. Allen[1], Mackenzie M. Grieman[2]

[1]British Antarctic Survey, Ice Dynamics and Paleoclimate, Cambridge, CB3 0ET, UK
[2]Department of Earth Sciences, University of Cambridge, Cambridge, CB2 3EQ, UK

*Correspondence to*: Dieter R. Tetzner (dietet95@bas.ac.uk)

**Abstract.** The Southern Hemisphere Westerly Winds are among the most important drivers of recently observed environmental changes in West Antarctica. However, the lack of long-term wind records in this region hinders our ability to assess the long-term context of these variations. Ice core proxy records yield valuable information about past environmental
changes, although current proxies present limitations when aiming to reconstruct past winds. Here we present the first regional wind study based on the novel use of diatoms preserved in Antarctic ice cores. We assess the temporal variability in diatom abundance and its relation to regional environmental parameters spanning a 20-year period across three sites in the southern Antarctic Peninsula and Ellsworth Land, Antarctica. Correlation analyses reveal that the temporal variability of diatom abundance from high elevation ice core sites is driven by changes in wind strength over the core of the Southern Hemisphere
Westerly Wind belt. Validating the use of diatoms preserved in ice cores from the Southern Antarctic Peninsula and Ellsworth Land as a proxy for reconstructing past variations in wind strength over the Pacific sector of the Southern Hemisphere Westerly Wind belt.

## 1 Introduction

Winds over the Southern Ocean (circumpolar westerlies) play a key role in driving the exchange of heat and carbon dioxide
between the ocean and atmosphere (Russell et al., 2006; Quéré et al., 2007; Hodgson and Sime, 2010; Landschutzer et al. 2015). In recent decades, the circumpolar wind belt has increased in strength and has shifted towards the Antarctic continent, constituting one of the strongest climatic trends in the Southern Hemisphere (Thompson and Solomon, 2002, Gille et al. 2008, Young et al. 2011). These atmospheric changes have been linked as drivers of the widespread warming observed in the Antarctic Peninsula (AP) (Orr et al., 2004; Van Den Broeke and Van Lipzig, 2004; Marshall et al., 2006; Thomas et al., 2009;
Thomas and Tetzner, 2018; Turner et al., 2020) and West Antarctica (WA) (Thomas et al. 2013) and as the mechanism behind the enhanced upwelling of deep and relatively warm, CO2-rich, oceanic water (Nakayama et al., 2018). The upwelling of circumpolar deep water has been shown to promote accelerated melting and thinning at the base of the ice shelves (Thoma et al., 2008; Jacobs et al., 2011; Pritchard et al., 2012; Steig et al., 2012; Wåhlin et al., 2013; Dutrieux et al., 2014; Favier et al., 2014; Gille et al., 2014; Joughin et al., 2014; Paolo et al., 2015; Holland et al., 2019), thus, threatening the stability of floating

ice shelves along the coastline of the Amundsen-Bellingshausen Seas and ultimately contributing to global sea-level rise (Pritchard et al., 2012). Changes in the circumpolar westerlies have also been linked to an observed pattern of increased snow accumulation in Antarctica over the 20th century, which has mitigated global mean sea level rise (Medley and Thomas., 2019). To assess the context of these recent wind changes, an extended record of wind strength is needed.

Direct meteorological observations in the AP region are sparse, relatively short-term and mostly constrained to coastal regions

(Lazzara et al., 2012; Oliva et al., 2017; Thomas and Tetzner, 2018; Turner et al., 2020). The scarcity of continuous long-term data can be partly addressed by using climate reanalyses datasets. These datasets comprise model outputs from simulations that are used to interpolate measured climate variables, providing accurate meteorological data in this region for the satellite-era (1979-present) (Tetzner et al., 2019; Dong et al., 2020; Zhu et al., 2021), but are not viable for studying pre-satellite regional climate variability. Climate models can be used to extend the record back in time, however, model simulations from the fifth

Coupled Model Intercomparison Project (CMIP5) exhibit strong biases over the Southern Ocean compared with observations (Bracegirdle et al., 2013). Overall, the lack of long-term wind records in the region hinders the possibility to assess the long-term context of the recently observed changes.

Ice cores provide valuable climatic information over a range of timescales (Legrand and Mayewski, 1997). Insoluble mineral dust concentrations in Antarctic ice cores have been widely used to infer past variability in regional-to-hemispheric

atmospheric circulation (Röthlisberger et al., 2002; Lambert et al., 2008; Koffman et al., 2014; Delmonte et al., 2020; Laluraj et al., 2020). Geochemical tracers of mineral dust, such as the non-sea-salt component of soluble calcium ($nssCa^{2+}$) or potassium ($nssK^+$), have been used to track changes in the strength and position of the Southern Hemisphere Westerly Wind (SHWW) belt (Dixon et al., 2012; Mayewski et al., 2013). However, these traditional proxies typically originate from South America, Australia and New Zealand and local Antarctic ice-free areas and are therefore potentially influenced by

environmental changes in these distal source regions (McConnell et al., 2007; Lambert et al., 2008; Li et al., 2010; Wolff et al., 2010; Bullard et al., 2016; Delmonte et al., 2017). Sea salt sodium ($ssNa^+$) has been previously interpreted as a proxy of increased meridional transport from the Southern Ocean to ice core sites in Antarctica (Kreutz et al., 2000; Goodwin et al., 2004; Kaspari et al., 2005; Mayewski et al., 2013; Vance et al., 2013). However, the $ssNa^+$ signal is potentially influenced by inputs from blowing snow above sea ice (Frey et al., 2020) and by the production of $ssNa^+$ in 'frost-flowers', as well as by

inputs from the open ocean (Huang and Jaeglé, 2017). As frost flowers are precipitated from the surface of freshly formed sea ice (Wagenbach et al., 1998; Rankin et al., 2000), and all these sources are affected by sea ice extent, some studies have reflected on the potential of the $ssNa^+$ ice core record from Antarctic coastal sites as a proxy of seasonal sea ice extension (Rankin et al., 2002; Wolff et al., 2003; Wolff et al., 2006). These conflicting sources of $ssNa^+$ suggest that the variability of the $ssNa^+$ record is not entirely driven by changes in winds or atmospheric circulation. The current limitations identified in

conventional ice core wind and atmospheric circulation proxies highlight the need to explore new species to reconstruct wind strength.

Diatoms entrained in Antarctic snow and ice have recently been proposed as an alternative source of information about winds (Allen et al., 2020). Diatoms are unicellular algae with siliceous cell walls that inhabit marine, brackish and freshwater

environments throughout the world (Smol and Stoermer, 2010). Despite their aquatic habitats, several studies support that they can be airborne (Chalmers et al., 1996; Gayley et al., 1989; Lichti-Federovich, 1984; McKay et al., 2008; Wang et al., 2008; Harper and Mackay, 2010; Spaulding et al., 2010; Hausmann et al., 2011; Budgeon et al., 2012; Papina et al., 2013; Fritz et al., 2015). Airborne diatoms can be sourced from both exposed diatom-bearing sediments and modern water bodies (e.g. oceans, streams, lakes) (Harper and McKay, 2010). Marine diatoms can be effectively lifted from the sea-surface microlayer into the atmosphere by wind-induced bubble-bursting and wave-breaking processes (Cipriano and Blanchard, 1981; Farmer et al., 1993). Once in the atmosphere, they can be transported by winds over long distances (Chalmers, 1996; Elster et al., 2007; Harper and McKay, 2010; Budgeon et al., 2012; Marks et al., 2019; Allen et al., 2020). Diatom records have been evaluated at multiple ice core sites in the AP and Ellsworth Land (EL) regions (Tetzner et al., 2021a) to reveal consistent spatial variations and increases in the diatom concentration during the past three decades. Despite the potential the diatom record has shown to reconstruct past wind strength, this proxy requires evaluation against more established ice core wind and atmospheric circulation proxies.

In this study, we evaluate the potential for diatoms to reconstruct regional wind strength using an array of three shallow depth ice cores drilled in the AP and EL regions. We investigate the relationship between diatom abundance and changes in environmental parameters to assess the regional consistency of the diatom-based wind proxy. We expand these analyses by comparing the diatom record to traditional ice core wind and atmospheric circulation proxies based on major ions and dust.

## 2 Methods

### 2.1 Ice core records and age scales

Three ice cores and firn cores from the southern AP and EL were included in this study (Figure 1) (Table 1). The Jurassic ice core (JUR, 140 m) was drilled in the vicinity of Jurassic Nunatak, using the BAS electromechanical drill during the austral summer 2012/2013. The Sky-Blu firn core (SKBL, 21.8 m) from the vicinity of Sky-Blu Field Station, southern AP, and the Sherman Island firn core (SHIC, 21.3 m) from Eights Coast, were drilled using a Kovacs hand-auger during the austral summer 2019/2020; hereafter, for practical reasons, firn cores will be referred to as ice cores. For JUR, SKBL and SHIC, an ice core chronology was established (Tetzner et al., 2021a) based on their hydrogen peroxide ($H_2O_2$) annual cycle that is assumed to peak during the austral summer solstice and to exhibit its minimum during the austral winter (Frey et al., 2006; Thomas et al., 2008). Ice core chronologies were corroborated using the annual cycle of the non-sea salt component of major ions, such as non-sea salt sulphate ($nssSO_4^{2-}$) (Piel et al., 2006), that is assumed to peak between November and January in this region (Pasteris et al., 2014; Thoen et al., 2018). Ice core chronologies were corroborated with the presence of volcanic tephra in the 2001 CE ice core layer (Tetzner et al., 2021b). The top 15 m of SKBL included in this work and the full SHIC core were dated back to 1999 CE, and the top 36.90 m of JUR included in this work dated back to 1992 CE. The estimated dating error is ±3 months for each year and with no accumulated error.

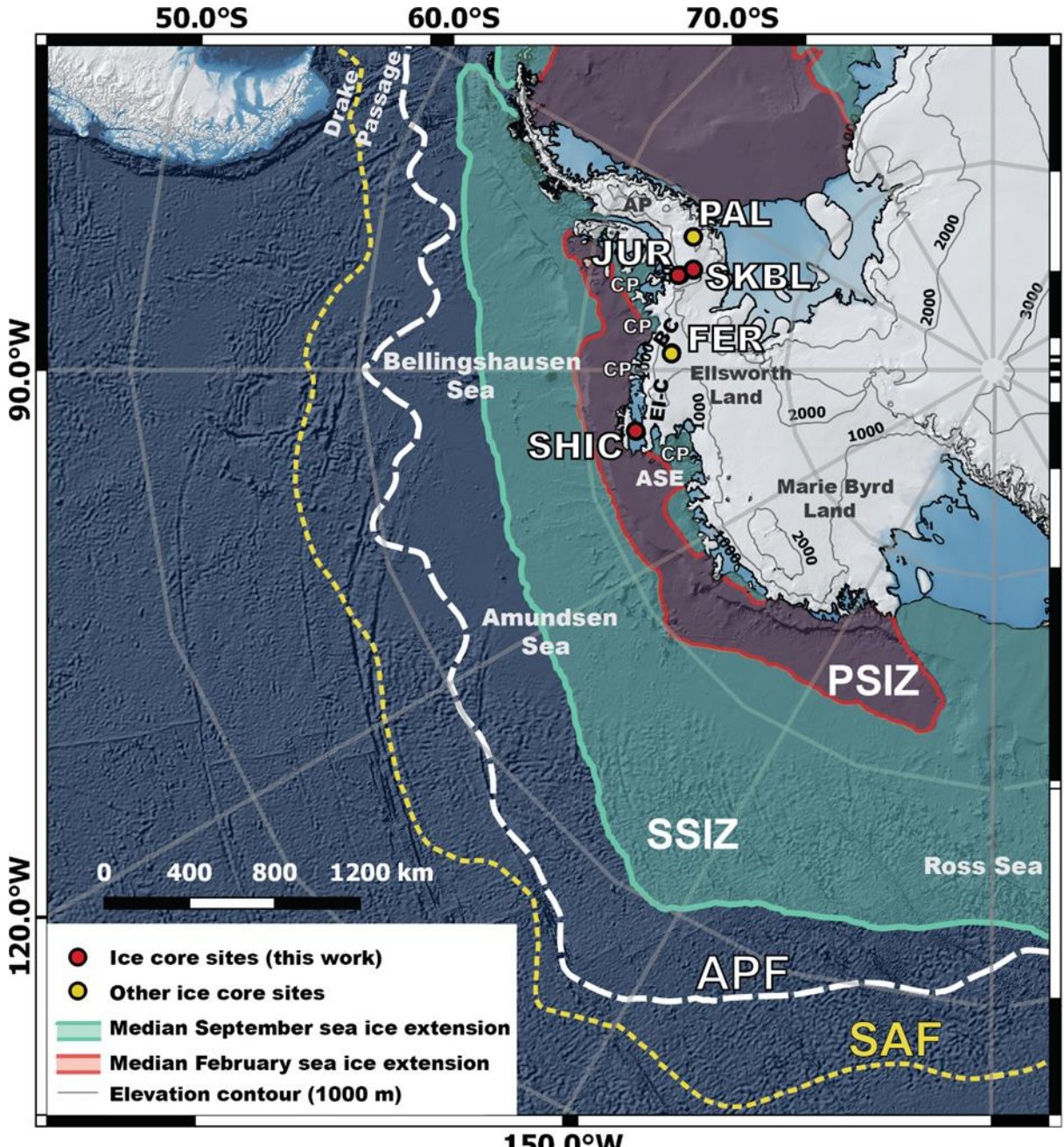

95

**Figure 1. Map showing the location of the ice core sites and main oceanographic features considered in this study. The red circles show the locations of the three ice core sites. The pale green line shows the median September sea ice-extension between 1980-2010 CE and the pale green area represents the Seasonal Sea ice Zone (SSIZ). The red line shows the median February sea ice-extension between 1980-2010 CE, and the pale red area represents the Perennial Sea ice Zone (PSIZ). SAF= Sub-Antarctic Front. APF= Antarctic Polar Front. CP= Coastal Polynya. EI-C= Eights Coast. BC= Bryan Coast. ASE=Amundsen Sea Embayment. PAL= Palmer ice core site. FER= Ferrigno ice core site. AP= Antarctic Peninsula.**

100

**Table 1. Summary of each ice core geographical location and main features analysed in this study. SIE= Sea Ice Edge(*) - The distance from SIE reported corresponds to the median for years covering the data interval. September SIE values used for calculations were obtained as distance between the ice core site and the closest point in the northern limit of 15% sea ice cover. February SIE values used for calculations were obtained as the distance between the ice core site and the closest sea ice free region.**

| Core name | Long | Lat | Elevation (m.a.s.l) | ANNUAL RECORD | | Total depth used (m) | Distance from SIE (km)* | |
|---|---|---|---|---|---|---|---|---|
| | | | | Years (CE) | # samples | | Sept | Feb |
| JUR | -73.06 | -74.33 | 1139 | 1992 - 2012 | 20 | 36.9 | 1045 | 140 |
| SKBL | -71.59 | -74.85 | 1419 | 1999 - 2019 | 20 | 15.0 | 1148 | 200 |
| SHIC | -99.63 | -72.67 | 474 | 1999 - 2019 | 20 | 21.3 | 753 | 130 |

## 2.2 Sample preparation and analyses

All ice cores included in this study were cut using a band-saw with a steel blade. Discrete ice core samples were cut at 5-cm resolution for ion chromatographic analyses using reagent-free Dionex ICS-2500 anion and IC 2000 cation systems in a class-100 cleanroom. Major ion concentrations were used to calculate the ionic contributions from marine and continental sources. The marine sea salt fraction of sodium (ssNa$^+$) and the non-sea salt fraction of calcium (nssCa$^{2+}$) were calculated using Equation (1) and Equation (2):

$$[ssNa^+] = [Na^+] - [nssCa^{2+}]/R_{crust} \tag{1}$$

$$[nssCa^{2+}] = [Ca^{2+}] - [ssNa^+]*R_{sea\ water\ Ca/Na} \tag{2}$$

Where R$_{crust}$ and R$_{sea\ water\ Ca/Na}$ are the mean ratios (weight by weight - w/w) of Ca/Na in the Earth crust (R$_{crust\ Ca/Na}$= 1.78) and in bulk sea water (R$_{sea\ water\ Ca/Na}$= 0.038), respectively (Bowen, 1979). The non-sea salt fraction of potassium (nssK$^+$) was calculated using the following Equation (3):

$$[nssK^+] = [K^+] - [ssNa^+]* R_{sea\ water\ K/Na} \tag{3}$$

Where R$_{sea\ water\ K/Na}$ is the mean ratio (weight by weight - w/w) of K/Na in bulk sea water (R$_{sea\ water\ K/Na}$= 0.036) (Piel et al., 2006).

Microparticle Concentration (MPC) was measured on each ice core using a flow-through Klotz Abakus laser particle counter connected to a continuous ice core melter system at the British Antarctic Survey (Grieman et al., 2021).

Diatom samples from each ice core site were previously analysed and reported by Tetzner et al. (2021a). All diatom samples were extracted by filtration. Ice core meltwater was filtered through polycarbonate membrane filters (pore diameter 1.0 µm) and subsequently scanned in a scanning electron microscope (SEM) following the method presented in Tetzner et al. (2021c). Observations of diatom preservation were based on the visual identification of characteristic frustule dissolution features and degradation described in Warnock and Scherer (2015). Diatom frustules and fragments less than 5 µm were excluded from counting and identification. Diatom counts per sample (n) included all diatom valves, partially obscured diatom valves and diatom fragments identified in each sample. After processing, diatom counts per sample (n) were represented as their temporal

equivalent, diatom abundance (n yr$^{-1}$). The diatom abundance parameter includes all diatoms and diatom remains identified on

each sample, regardless of their potential source. As reported by Tetzner et al. (2021a), the diatom assemblages at all ice core sites presented in this study is primarily comprised by *Fragilariopsis cylindrus*, *Fragilariopsis pseudonana*, *Pseudonitzschia* spp., *Shionodiscus gracilis* and *Cyclotella* group., with the exclusive presence of other diatom species on each site (Appendix A – Table A1).

All correlations are based on linearly detrended data and calculated using the Pearson's linear correlation (R). Time series

linear correlations were calculated over a 20-year period (1992-2012 CE for JUR and 1999–2019 CE for SHIC and SKBL). Seasons are reported as austral summer (December to February, DJF), autumn (March to May, MAM), winter (June to August, JJA) and spring (September to November, SON).

## 2.3 Climate reanalyses and sea ice extension data

Monthly reanalysis fields from the fifth generation of the European Center for Medium-Range Weather Forecasts (ECMWF),

ERA5 (Hersbach and Dee, 2016), were used to obtain spatial correlations between ice core records and environmental parameters. Three fields were used to perform spatial correlation analyses: wind speed (10 m wind speed), precipitation and sea ice cover. ERA5 datasets provide hourly data available at 0.25-degree resolution (~31 km) since 1979 CE.

Sea ice extension data were obtained from the Sea Ice Index, Version 3 dataset (Fetterer et al., 2017) from the National Snow and Ice Data Centre (NSIDC), providing monthly sea ice concentrations at 25 km resolution from 1979 to 2021 CE. September

sea ice limits (defined as the median northerly extent of 15% sea ice cover) were considered the annual sea ice maximum and February sea ice limits (defined as the median northerly extent of 15% sea ice cover) were considered the annual sea ice minimum (Thomas et al., 2019).

## 2.4 Statistical analyses

For each ice core site, the diatom abundance was compared to conventional ice core wind and atmospheric circulation proxies

(ssNa$^+$, nssCa$^{2+}$, nssK$^+$ and MPC) and to records of methane sulphonic acid (MSA), a traditional sea ice proxy. All correlations were based on annual averages (winter-to-winter). The correlation of chemical, MPC and diatom abundance records across all ice core sites was carried out using the flux parameter (except for diatom abundance). Annual fluxes were calculated after multiplying the annual mean concentration by the estimated annual snow accumulation (mm of water equivalent). Spatial correlations were calculated using chemical and MPC mean annual concentrations and diatom abundance. Spatial correlations

were generated using the field correlation tool from the Royal Netherlands Meteorological Institute (KNMI) Climate Explorer (https://climexp.knmi.nl/start.cgi). Spatial correlations were based on linearly detrended data and calculated using the Pearson's linear correlation (R) over the South Pacific sector, from 180°W to 60°W and 40°S to 90°S, selected based on air-mass pathways reaching the ice core sites (Thomas and Bracegirdle, 2015). Based on the records of September sea ice cover between 1992-2020 CE (Fetterer et al., 2017), spatial correlations for sea ice cover were calculated over a reduced region

(between 180°W to 60°W and 55°S to 90°S). All linear correlations were calculated over a 20-year period (1992-2012 CE for

JUR and 1999–2019 CE for SHIC and SKBL). The MPC record from JUR covers the interval 1992-2011 CE, therefore, correlations calculated using this parameter at JUR are calculated over a 19-year period. Since diatom concentrations at SHIC peak during the austral summer (Tetzner et al., 2021a), spatial correlations for the SHIC diatom record were also calculated over 6-month period covering the austral spring & summer season (September-February).

## 3 Results

### 3.1 Jurassic ice core (JUR)

#### 3.1.1 Chemistry, MPC and diatom abundance annual records

The JUR chemical fluxes, MPC flux and diatom abundance records are characterized by positive trends during the 1992-2012 CE period, with the exception of MSA flux which exhibits a negative trend (Figure 2) (Appendix B – Table B1). Of them, only $nssCa^{2+}$flux and $nssK^+$ flux were statistically significant ($p<0.05$) (Appendix B - Table B1). The $nssCa^{2+}$ flux shows a steady increase, reaching a maximum between 2010-2011 CE. The $nssK^+$ flux shows a similar pattern with values increasing steadily after 2002 CE. The $ssNa^+$ flux is moderately variable from 1992 to 2012 CE, with an overall increase between 2000 and 2002 CE. The MSA flux displays an overall decrease between 1992-2003 CE, followed by a general increase during 2003-2012 CE. Similarly, the MPC exhibits an increase during 1992-2012 CE with its lowest values during 1998 CE. The diatom abundance is moderately variable between 1992-2004 CE, followed by a marked increase after 2004 CE, reaching its highest value during 2009-2010 CE (Figure 2). Similar features were present in the JUR chemical and MPC concentration records (Appendix C – Table C1, and Appendix C – Figure C1).

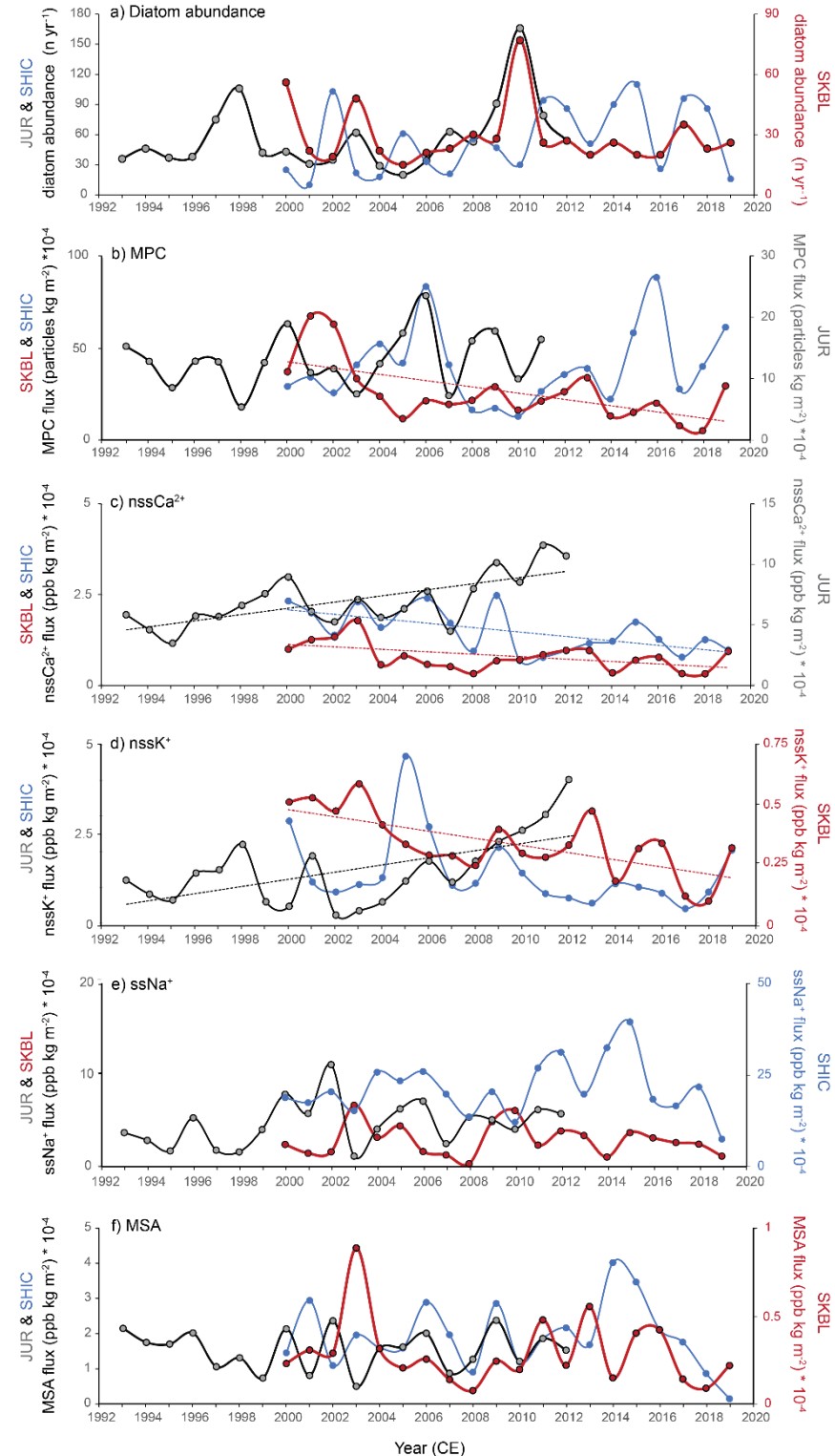

### 3.1.2 Environmental correlations

All records (excluding MSA) presented strong and significant correlations with wind strength (R≤-0.6 & R≥0.6, p<0.05) (Appendix D – Figure D1)). Diatom abundance is positively correlated with wind strength (0.45≤R≤0.67, p<0.05) over a latitudinal band at the northern limit of the Amundsen Sea (~60°S), and negatively correlated with the sea ice cover north of the SHIC ice core drilling site (-0.45≥R≥-0.69, p<0.05) (Figure 3). The MPC is positively correlated with wind strength over the east coast of Argentina (0.6≤R≤0.73, p<0.05). The $nssCa^{2+}$ exhibits a positive correlation with wind strength (0.6≤R≤0.62, p<0.05), and with precipitation (0.6≤R≤0.77, p<0.05), off the western coast of southern South America. The $nssK^+$ shows a positive correlation with wind strength (0.6≤R≤0.66, p<0.05) over the Amundsen Sea. The $ssNa^+$ is positively correlated with wind strength over the Drake Passage (0.6≤R≤0.7, p<0.05) and over Ellsworth Land (0.6≤R≤0.63, p<0.05). No significant correlations were identified with wind strength or precipitation at, or near, the JUR ice core drilling site. No strong correlations (R≥0.6 or R≤-0.6) between the MSA record and sea ice cover were identified in the region.

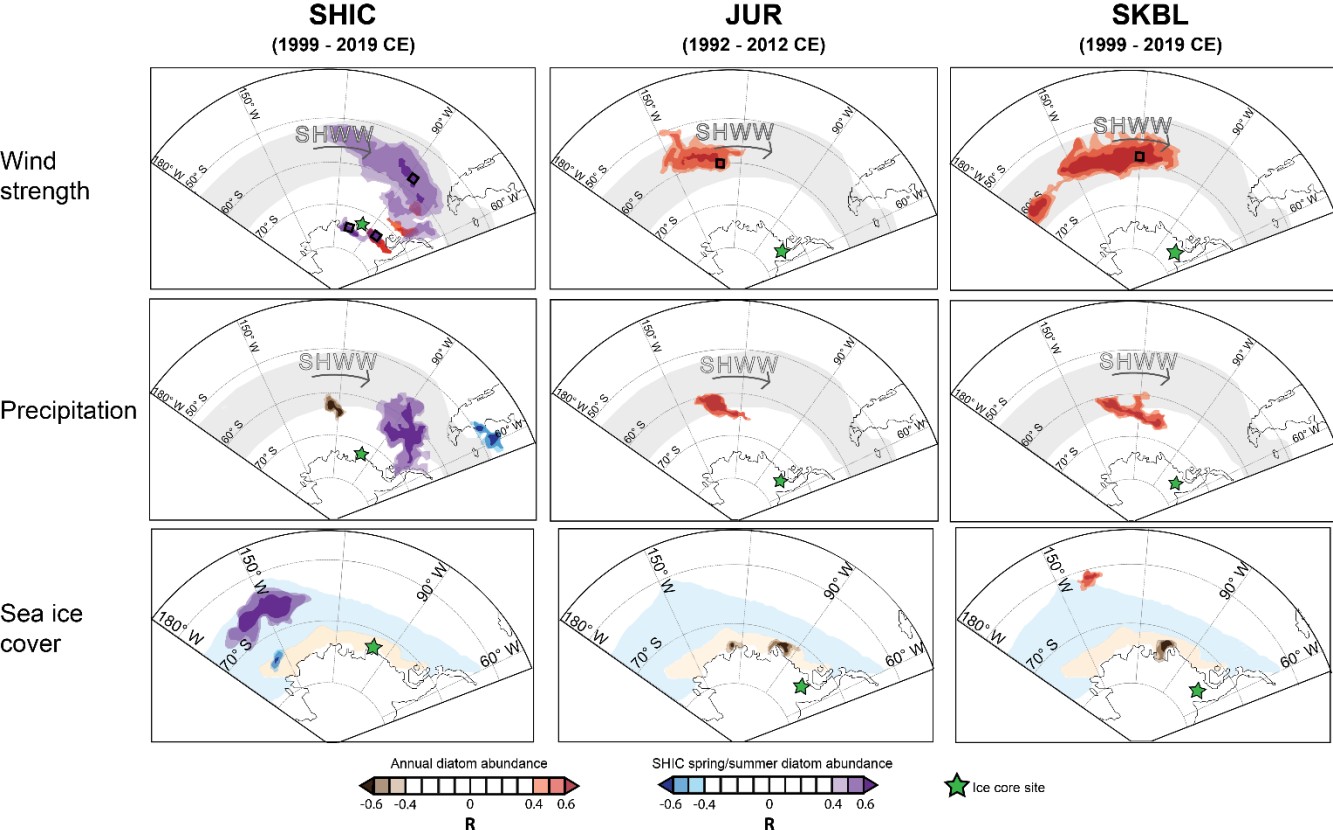

 **Figure 3. Regional maps showing spatial correlations between environmental parameters and diatom abundance. Colour-coded polygons in the maps indicate highly correlated regions (R≥0.45 or R≤-0.45). All polygons plotted are statistically significant (p<0.05). Pale grey latitudinal band indicates the core of the Southern Hemisphere Westerly Wind (SHWW) belt. Pale blue polygons indicate the region covered by seasonal sea ice. Pale orange polygons indicate the region covered by perennial sea ice. Black squares represent a 1°x1° quadrant around the point of highest correlation (QHC) between wind strength and diatom abundance. Regional maps only present areas of spatial correlation larger than 1°x1°. Degrees of freedom (df) for each spatial correlation can be obtained using the following expression dependent on the sample size (n); df=n-2.**

### 3.2 Sky-Blu ice core (SKBL)

### 3.2.1 Chemistry, MPC and diatom abundance annual records

The SKBL chemical fluxes, MPC flux and diatom abundance records are characterized by negative trends during the 1999-2019 CE period (Figure 2) (Appendix B – Table B1). Among these trends, only MPC, $nssCa^{2+}$ and $nssK^+$ fluxes exhibited linear trends that were statistically significant (p<0.05). The diatom abundance is characterized by moderate interannual variability and single major increases during 1999-2000, 2002-2003 and 2009-2010 CE. The later presenting the absolute highest value (77 n $yr^{-1}$). A major increase in MSA flux was also registered during 2003 CE (Figure 2). Similar features were present in the SKBL chemical concentration and MPC records (Appendix C – Table C1, and Appendix C – Figure C1).

### 3.2.2 Environmental correlations

The diatom abundance, chemical and MPC records from SKBL were compared to environmental parameters over the 1999-2019 CE period (Appendix D – Figure D1). Two records are highly correlated with wind strength, the diatom abundance and $ssNa^+$. Diatom abundance exhibits a positive correlation with wind strength (0.45≤R≤0.78, p<0.05) over a latitudinal band (~60°S) extending from the Ross Sea to the Amundsen Sea and a negative correlation with the sea ice cover (-0.6≥R≥-0.71, p<0.05) over the Amundsen Sea Embayment (Figure 3). The $ssNa^+$ is positively correlated with wind strength (0.6≤R≤0.75, p<0.05) off the west coast of South America at ~45°S. No correlations were identified between the ice core records and wind strength or precipitation over the SKBL ice core drilling site. No strong correlations (R≥0.6 or R≤-0.6) between the MSA record and sea ice cover were identified in the region.

### 3.3 Sherman Island ice core (SHIC)

### 3.3.1 Chemistry, MPC and diatom abundance annual records

The SHIC chemical fluxes, MPC flux and diatom abundance records present contrasting features over the 1999-2019 CE period (Figure 2) (Appendix B – Table B1). Negative linear trends are observed in $nssCa^{2+}$ flux, $nssK^+$ flux and MSA flux. The $ssNa^+$ flux exhibits a positive trend with values reaching an absolute maximum during 2015 CE. The MSA flux was relatively constant over the 1999-2014 CE period. After 2014 CE, MSA flux decreases, reaching its absolute minimum between 2018-2019 CE. The MPC flux shows a steady increase after 2008-2009 CE, reaching its highest value in 2018-2019 CE. The diatom abundance exhibits a positive trend (2.16 n $yr^{-1}$, p>0.05) with a strong interannual variability which describes a semi-

periodic 3-year pattern (Figure 2). Similar features were present in the SHIC chemical concentration and MPC records (Appendix C – Table C1, and Appendix C – Figure C1).

### 3.3.2 Environmental correlations

The diatom abundance, chemical and MPC records from SHIC were compared to environmental parameters over the 1999-2019 CE period (Appendix D – Figure D1). Diatom abundance, MPC and $ssNa^+$ are highly correlated with wind strength over the 1999-2019 CE period. Annual diatom abundance is positively correlated with wind strength ($0.45 \leq R \leq 0.61$, $p<0.05$) over the Bryan Coast sector (Figure 3). The spring/summer diatom abundance is positively correlated with wind strength over a latitudinal band at the northern limit of the Bellingshausen Sea (~60°S) ($0.45 \leq R \leq 0.64$, $p<0.05$), over the Bryan Coast

($0.45 \leq R \leq 0.64$, $p<0.05$) and over the Amundsen Sea Embayment ($0.45 \leq R \leq 0.71$, $p<0.05$) (Figure 3). In parallel, the spring/summer diatom abundance is positively correlated with the sea ice cover over the Amundsen Sea ($0.45 \leq R \leq 0.81$, $p<0.05$) (Figure 3). MPC is positively correlated with precipitation ($0.6 \leq R \leq 0.66$, $p<0.05$) over the SHIC drilling site. Conversely, MPC is negatively correlated with precipitation off the southern coast of South America ($-0.6 \geq R \geq -0.65$, $p<0.05$). The $ssNa^+$ is positively corelated with sea ice ($0.6 \leq R \leq 0.7$, $p<0.05$) to the west and north-west of the SHIC drilling site. No clear correlations

were identified with the $nssCa^{2+}$ and $nssK^+$ records. No strong correlations ($R \geq 0.6$ or $R \leq -0.6$) between the MSA record and sea ice cover were identified for the SHIC site.

### 3.4 Correlation of ice core records

Linear correlations were calculated between the annual chemical and MPC fluxes, and diatom abundance records across all ice core sites (Supplementary material – Table S1). Strong positive and statistically significant ($p<0.05$) correlations were

245 identified between $nssCa^{2+}$ flux and $nssK^+$ flux in SHIC, JUR and SKBL ($R=0.48$, $R=0.45$ and $R=0.82$, respectively) and between $ssNa^+$ flux and MSA flux in SHIC, JUR and SKBL ($R=0.7$, $R=0.63$ and $R=0.52$, respectively). Diatom abundance and $ssNa^+$ flux exhibited strong and significant correlations ($p<0.05$) across all the ice core sites ($R=0.6$, $R=-0.47$ and $R=0.46$, for SHIC, JUR and SKBL, respectively). The only statistically significant correlation in the same proxy across the sites was between diatom abundance in the JUR and SKBL cores ($R=0.84$, $p<0.05$). No clear or consistent pattern was identified when

comparing chemical proxies from different ice core sites.

## 4 Discussion

### 4.1 Temporal variability of the diatom record

The diatom abundance preserved in ice cores reveals a strong interannual variability across the AP and EL regions from 1992-2019 CE. The relationship between diatom abundance and environmental parameters at the ice core drilling sites are weak and

255 not statistically significant. Indicating that conditions at the ice core site are not drivers of the diatom temporal variability, in

agreement with previous studies concluding that ice core diatom records are not dependent on annual snow accumulation or local wind conditions (Allen et al., 2020; Tetzner et al., 2021a).

Spatial correlations of annual wind strength and the annual diatom abundance preserved in ice cores reveal regions of significant positive correlations ($0.45 \leq R \leq 0.78$, $p < 0.05$) (Figure 3). These regions match well with the oceanographic zones indicated by the ecology of the dominant marine diatom taxa (Tetzner et al., 2021a). Furthermore, the neighbouring ice core sites of JUR and SKBL share similar regions of spatial correlation ($0.45 \leq R \leq 0.78$, $p < 0.05$) (Figure 3). Numerous studies have highlighted the relationship between wind strength and sea-spray production (Blanchard, 1963; Schlichting, 1974; Monahan et al., 1983; Callaghan et al., 2007; Löndahl, 2014; Tesson et al., 2016; Wiśniewska et al., 2019; Marks et al., 2019). Stronger winds have been shown to enhance the production of sea-spray aerosols, including microalgae, which implies an increased transfer of diatoms from the sea-surface microlayer into the atmosphere (Marks et al., 2019). Studies on regional atmospheric circulation confirm that air masses from these regions are effectively transported to ice core sites (Thomas and Bracegirdle, 2009; Abram et al., 2010; Thomas and Bracegirdle, 2015; Allen et al., 2020), therefore establishing an efficient transport pathway from the identified source regions to the ice core sites.

Two oceanographic zones were highlighted in the spatial correlation analyses for their strong link between changes in wind strength and changes in diatom abundance; the seasonal sea ice zone (SSIZ) and permanently open ocean zone (POOZ). The JUR and SKBL diatom records were correlated exclusively with winds in the POOZ (northern Amundsen Sea) and the SHIC spring/summer diatom record was correlated with regions from both the SSIZ (Bryan Coast sector & Amundsen Sea Embayment) and POOZ (northern Bellingshausen Sea) (Figure 3). It is important to note that these regions of spatial correlation agree with the oceanographic affiliations of the diatom assemblages for the respective ice cores (coastal vs inland sites) reported by Tetzner et al. (2021a). Despite the correlation between the SHIC spring/summer diatom record and winds in the POOZ, the diatom assemblages for SHIC support a stronger link with the SSIZ (Tetzner et al. 2021a). Therefore, discussion for the SHIC diatom record will be focused on environmental changes within the SSIZ.

### 4.1.1 SSIZ controls on diatom variability at a coastal site (SHIC)

The SSIZ is identified as the dominant source region by both the spatial correlation analyses and the ecological associations of the SHIC diatom record (Tetzner et al., 2021a). Wind strength and sea ice dynamics play an important role in driving the primary productivity in diatom source regions within the SSIZ (Arrigo et al., 2008; Arrigo et al., 2012). In the Bryan Coast and Amundsen Sea Embayment, winds drive sea ice northward producing large ice-free areas (polynyas) near the coast (Arrigo et al., 2012; Holland and Kwok, 2012). These coastal polynyas have been identified as among the most productive regions in the Southern Ocean (Arrigo et al., 2008; Soppa et al., 2016). The stronger link to wind strength during the austral spring and summer (Figure 3), and strong seasonality identified in the sub-annual diatom record reflect the enhanced austral spring/summer primary productivity in these regions (Arrigo et al., 2008; Soppa et al., 2016; Tetzner et al., 2021a). This suggests that the interannual diatom variability at this site is not exclusively driven by wind strength, but also dependent on sea ice dynamics and/or primary productivity. Previous studies have identified recent reductions in the ice-covered days (sea

ice concentration) during the austral summer (Stammerjohn et al., 2012) and the development of coastal polynyas (Eltanin
Polynya, Pine Island Polynya and Amundsen Sea Polynya) within the regions identified as the SHIC diatom sources (Arrigo
and van Dijken, 2003; Arrigo et al., 2012). Both lead to prolonged and increased biogenic primary productivity near the ice
core site. Greater sea ice breakup and more open water in summer often result in larger phytoplankton blooms (Soppa et al.,
2016). A prolonged ice-free diatom source and/or recently enhanced primary productivity will increase the availability of
diatoms on the ocean surface, favouring potentially larger entrainments of diatoms by strong winds. This is consistent with the
recent large decadal increases seen in the SHIC diatom concentration (Tetzner et al., 2021a).

The clear link identified between the seasonality of the diatom record and sea ice primary productivity is complemented by
the strong relationship between the austral spring/summer diatom abundance and sea ice cover over the Amundsen Sea.
However, no direct relationship was observed between the diatom abundance and the sea ice cover directly in the vicinities of
SHIC (Figure 3). This is consistent with the lack of correlation between MSA and sea ice at this site, despite previous studies
observing a positive relationship between MSA and sea ice extent in this region (Abram et al., 2010; Thomas and Abram,
2016). These findings suggest that diatom (and MSA) variability at SHIC is not exclusively driven by either marine primary
productivity or changes in sea ice cover. This can be explained by the effect of a bi-directional pattern of winds at SHIC,
drawing air masses from both the east (Bryan Coast) and the west (Amundsen Sea Embayment) (Tetzner et al., 2019) resulting
in two distinct sources of aerosols to the ice core site. Additionally, the onset of the sea ice breakup and duration of ice-free
conditions at the Bryan Coast, Eights Coast and the Amundsen Sea Embayment does not follow a clear pattern. Some years
present these regions completely ice-free, while others present either one or more regions ice-covered (Arrigo et al., 2008).
The development of coastal polynyas near SHIC (See Figure 1) will contribute to the increased availability of diatoms in some
years (Arrigo and van Dijken, 2003; Arrigo et al., 2015). The potential input of diatoms from sources from opposite directions,
along with the variability in the distribution and duration of the sea ice cover and the occasional development of coastal
polynyas adds an additional and non-negligible source of variability to the diatom record at SHIC.

### 4.1.2 POOZ controls on diatom variability at inland sites (JUR and SKBL)

The diatom source region identified by the spatial correlation analyses and diatom ecological affiliations for JUR and SKBL
is in the northern Amundsen Sea within the POOZ (Figure 1 and Figure 3). This oceanographic zone is characterized by the
lowest seasonal and interannual variability in primary productivity for the whole SO (Arrigo et al., 2008; Soppa et al., 2016).
The absence of sea ice in the POOZ reduces the seasonal extremes in primary productivity evident within the SSIZ (Tetzner
et al., 2021a). The diatom availability in northern Amundsen Sea POOZ presents a low variability year-round (Arrigo et al.,
2008; Soppa et al., 2016), limiting the impact of interannual changes in primary productivity on the diatom record and
upholding that temporal variability in the JUR and SKBL diatom records is derived mainly from changes in wind strength at
the source. The marine source region supplying diatoms to JUR and SKBL is in the core of the SHWW belt. The significant
positive correlation between wind strength and annual diatom abundance ($0.45 \leq R \leq 0.78$, $p<0.05$) demonstrate the diatom

records from these sites can be used as a proxy for interannual changes in the strength of the SHWWs. However, changes in ocean primary productivity may contribute to diatom variability over longer time-scales.

Results presented here are consistent with the relationship identified between SHWWs strength and diatoms in an EL ice core (Allen et al., 2020). Specifically, our results show a close agreement with the location (50-60°S, 140°W) and magnitude of the strongest correlation (R=0.61, p<0.05) between annual SHWW anomalies and calibrated diatom flux at the Ferrigno ice core site (Figure 1) (Allen et al., 2020). Unlike JUR and SKBL, the Ferrigno diatom assemblage was characterised by the absence of sea ice diatoms. This can be explained by the more north-westerly location of the Ferrigno diatom source (compared to the JUR and SKBL source region) that is less likely to receive inputs from the SSIZ. Additionally, the absence of sea ice diatoms at Ferrigno supports the conclusion that inland sites have limited influence from low-elevation airmasses in contact with the SSIZ (Tetzner et al., 2021a).

### 4.1.3 Recent regional environmental changes and the diatom record

Ice core diatom records have the potential to capture regional environmental changes. The correlation between the annual diatom abundance variability at JUR and SKBL with changes in the wind strength in the SHWW belt (Figure 3) (Figure 4a and 4b) strongly suggests that the recent increase in diatoms at these ice core sites (Tetzner et al., 2021a) is driven by a strengthening of the SHWW belt during the satellite-era (Mayewski et al., 2013; Young and Ribal, 2019; Goyal et al., 2021). This link is further supported by sea-spray production experiments showing that a ~10% increase in wind strength ($U_{10}$>5 m s$^{-1}$) has the potential to double the production of spume drops and sea-spray aerosol, including microalgae (Monahan et al., 1986; Wu, 1993; Anguelova et al., 1999). Increasing the entrainment of aerosols into the atmosphere would likely increase the supply of diatoms to the ice core sites and therefore the number of diatoms contained in the ice. Despite SHIC exhibiting a similar increase in annual diatom abundance, the intrinsic relationship between the SHIC diatom record, the sea ice cover in SSIZ, and winds over the SSIZ prevents establishing a direct link between the increases in annual diatom abundance at SHIC and the strengthening of the SHWW belt (Figure 4c).

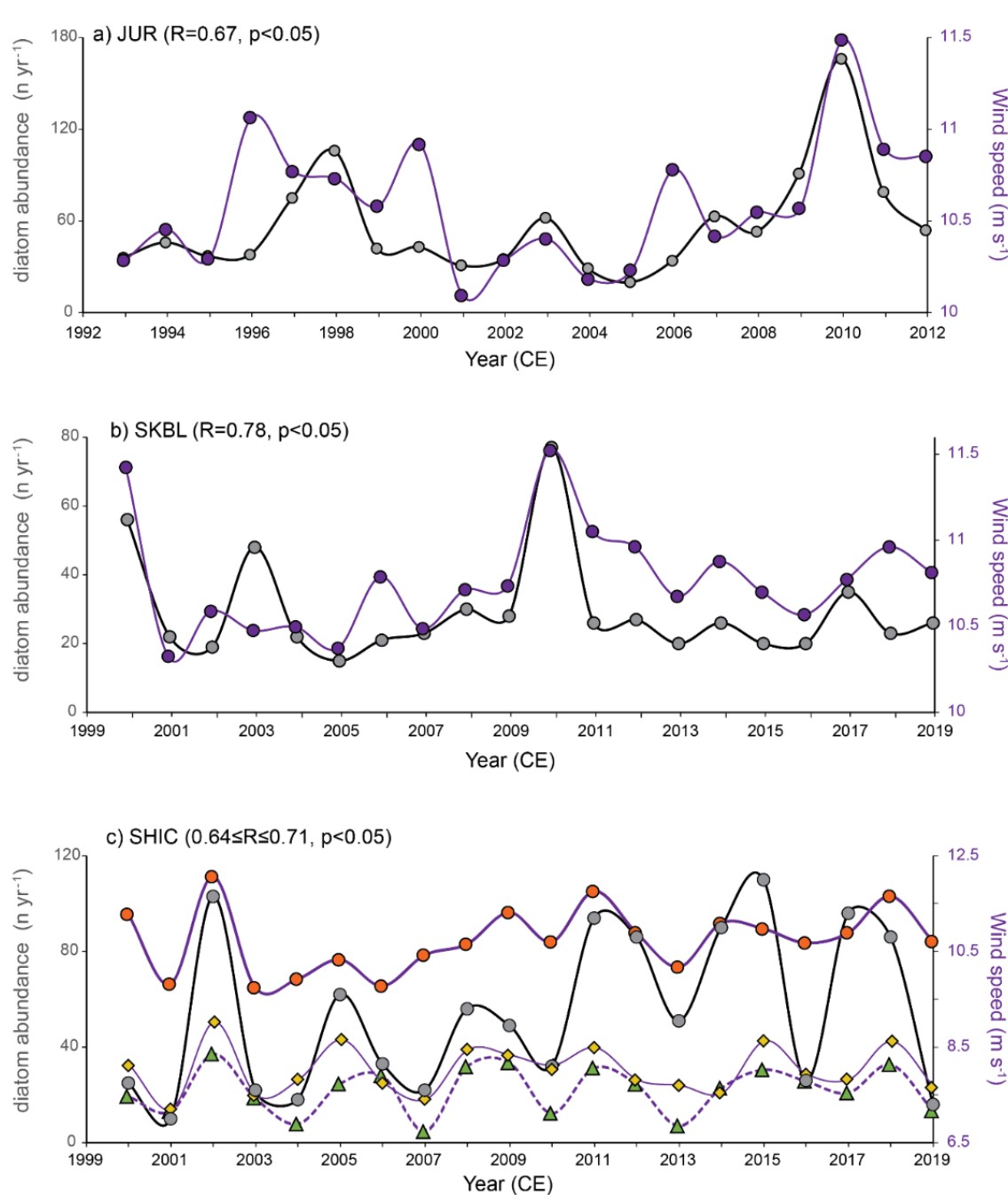

**Figure 4.** Time series of diatom abundance and mean wind speed inside the quadrants of highest correlation (QHC) for each ice core site highlighted in Figure 3. Data points represent the annual austral winter-to-winter average and were plotted over the correspondent austral summer. Colour-coded symbols for SHIC ice core represent the time series of wind speed on each of the QHC identified for that site. QHC-BS= Quadrant of high correlation over the Bellingshausen Sea. QHC-ASE= Quadrant of high correlation over the Amundsen Sea Embayment. QHC-BC= Quadrant of high correlation over the Bryan Coast.

## 4.2 The diatom wind proxy and the traditional ice core wind and atmospheric circulation proxies

We have demonstrated that the annual diatom abundance shows promise as a wind proxy of SHWW at multiple sites. To assess the full scope and validity of this novel proxy, we compared the new diatom records with established ice core wind and atmospheric circulation proxies.

MPC, $nssCa^{2+}$, $ssNa^+$ and $nssK^+$ have been used to infer past variability in regional-to-hemispheric atmospheric circulation from annual to millennial timescales. Our results reveal the limitations in these traditional proxies, which fail to reproduce the

interannual variability of wind strength upwind from the ice core sites. Even though many of these records show a strong link with wind strength, their region of correlation did not fit with their expected sources and were not consistent between sites. This refutes a direct link between wind strength and their in-situ aerosol production. For example, there are strong correlations between continental-ions and wind strength over regions in the middle of the ocean, where it is not feasible to entrain continental-ions (JUR $nssK^+$ record). Similarly, some records are strongly linked to other environmental parameters such as

precipitation, suggesting that source (JUR $nssCa^{2+}$ record) or site conditions (MPC record at SHIC), rather than wind strength, dominate the variability of these species. Some proxies (MPC at JUR and $ssNa^+$ at SKBL) were strongly linked to wind strength over regions north (~40-45°S) of the core of the SHWW belt (Appendix D – Figure D1). These results suggest that those records could indicate changes in wind strength over continental or marine regions, far from the core of the SHWW belt. Previous studies have demonstrated MPC and chemical species are valuable records to infer past changes in regional-to-

hemispheric wind regimes and atmospheric circulation. However, it must be acknowledged the variability of these tracers represents the cumulative effect of several factors (McConnell et al., 2007; Lambert et al., 2008; Li et al., 2010; Wolff et al., 2010; Bullard et al., 2016; Delmonte et al., 2017). These include (1) wind strength at the source(s), (2) airmass transport pathways, (3) primary production of the tracer at the source(s), (4) the humidity/precipitation while transported in the atmosphere and, (5) snow accumulation rate in Antarctica. The numerous factors influencing the variability of these records

hinder their capacity to accurately represent changes in winds strength. Despite our results confirm these tracers are not the ideal records to infer past wind strength variability, they do not discard their use to reconstruct wind and atmospheric circulation changes in a broad sense. Additionally, our results highlight the need for a thorough site and proxy evaluation before using these chemical and MPC species to reconstruct wind changes through time.

The $ssNa^+$ flux was the only record with a regionally consistent relationship with the diatom abundance (yellow cells in

Supplementary material - Table S1). Despite the strong correlations at each ice core site, they presented opposite signs (R<0 for JUR & R>0 for SHIC and SKBL). Suggesting that the interannual variability of diatom abundance and $ssNa^+$ flux is not driven exclusively by the same environmental parameters across the AP and EL region.

Correlation analyses of ice core proxies between sites confirm that diatom abundance is the only record with a consistent, statistically significant and strong correlation (JUR vs SKBL, R=0.84, p<0.05) (grey cell in Supplementary material - Table

S1), supporting the regional validity of the diatom-based wind proxy at the high elevation sites.

Previous studies have proposed MPC as a proxy for winds and atmospheric circulation in the AP and EL region (Mosley-Thompson et al., 1990; Mosley-Thompson et al., 1991; Thompson et al., 1994), suggesting that the MPC record is indicative of long-term regional environmental changes. However, low background concentrations reported in those studies suggest the MPC annual record from the AP and EL region could be biased by the occurrence of sporadic major dust events reaching the high southern latitudes (Li et al., 2010). Our results are in close agreement, showing that the regional MPC record preserved in ice cores from the AP and EL is not a robust indicator of interannual wind strength variability, possibly due to the numerous factors driving MPC variability. In contrast, the WAIS Divide ice core in neighbouring Marie Byrd Land, revealed a link between the coarse particle percentage (analogous to MPC) and interannual zonal wind strength variability at both 700hPa and 850hPa over the satellite-era (R=0.3-0.5, p<0.01) (Koffman et al., 2014). This contrasting behaviour can be partly explained by the diametrically opposed continental regions supplying mineral dust to Antarctica (Sudarchikova et al., 2014; Neff and Bertler, 2015) and the higher dependence of Antarctic Peninsula sites on extreme precipitation events (Turner et al., 2019). Similarly, differences in transport pathways and the role of local meteorology have also been used to explain the different behaviour observed in sea ice proxies (MSA) around Antarctica (Abram et al., 2013; Thomas et al., 2019). Our findings provide further evidence that ice core proxies perform differently depending on the geographical region and the temporal resolution over which they are assessed.

The temporal variability of chemical tracers (nssCa$^{2+}$, nssK$^+$, ssNa$^+$) in a network of ice cores across Marie Byrd Land (ITASE) have been proposed as tracers of long-term (multi-year) changes in atmospheric circulation (Dixon et al., 2012; Mayewski et al., 2013). However, only nssCa$^{+2}$ proved reliable in capturing interannual variability in atmospheric circulation patterns over the satellite-era (0.37≤R≤0.59, p<0.01), although not directly related to wind strength, but to northerly air mass incursions (Dixon et al., 2012). Thus, there is clearly a need for a reliable proxy for past wind strength in this important region of Antarctica. Based on our results, we propose that diatoms show the most promise as a tool to reconstruct the interannual variability of SHWW strength.

### 4.3 Recommendations

Our findings demonstrate that diatoms show the most promise as a proxy for past wind strength in AP and EL ice cores when compared with a range of traditional chemical and MPC records. At inland sites, diatom abundance is most strongly dependent on annual changes in the strength of the SHWWs and has excellent potential as a proxy to reconstruct past changes in the strength of the SHWWs over the northern Amundsen Sea.

Based on our results, we propose three candidate ice core sites across the AP and EL region where multi-decadal to centennial scale reconstructions of winds could be achieved: the Jurassic, Ferrigno and Palmer ice cores (Thomas and Bracegirdle, 2015) (See Figure 1). Evaluations have already confirmed the suitability of the JUR site (this study) and the Ferrigno site (Allen et al., 2020) in reproducing recent changes in the SHWW strength. Previous studies on the Ferrigno ice core have confirmed that this ice core covers the last three centuries (Thomas et al., 2015; Thomas and Abram, 2016), making it a good candidate for extending a SHWW reconstruction back to the core of the Little Ice Age (~1700 CE), while the JUR site has the potential to

capture changes over the last 140-years (Emanuelsson et al., 2022). The diatom record from the Palmer ice core has not yet been explored, however, this high elevation, low snow accumulation site (Thomas and Bracegirdle, 2015) has the potential to hold a 390-year record (Emanuelsson et al., 2022), one of the longest records in the region. An annually-resolved wind strength reconstruction from any of these three ice cores will establish if the recent intensification of the SHWW belt is unprecedented over the last few centuries and will clearly identify the timing of onset (Abram et al., 2013; Koffman et al., 2014; Turney et al., 2016; Perren et al., 2020).

Despite the consistency of the observations presented in this work, it must be acknowledged that they are from a particular region in Antarctica. Antarctica is a vast continent with contrasting environmental conditions in different sectors (Jones et al., 2016; Stenni et al., 2017; Thomas et al., 2017; Thomas et al., 2019). Thus, we recommend conducting a detailed assessment of the diatom proxy outside the AP-EL regions.

## 5 Conclusions

Ice cores capture changes in past wind strength and atmospheric variability, with several chemical and microparticles suggested as potential proxies. In this study, we propose an exciting new wind proxy, based on diatoms, that performs better than traditional ice core wind proxies in ice cores from the Southern Antarctic Peninsula and Ellsworth Land. A set of ice cores from this region have been analysed to demonstrate that the temporal variability of the diatom record preserved in inland ice cores is exclusively driven by changes in the wind strength within the core of the Southern Hemisphere Westerly Wind belt. While the temporal variability of the diatoms preserved in coastal ice cores is driven by a combination of wind strength and sea ice dynamics within the seasonal sea ice zone.

The lack of coherence between ice core chemical and dust proxies at sites from the same region suggest that these proxies should be used with caution. For these proxies, both the production at the source and the deposition at the ice core sites drive the variability, making it hard to resolve the signal of wind strength. These findings demonstrate the importance of a thorough site evaluation before extending environmental interpretations back in time, with no single proxy suitable for reconstructions across all of Antarctica.

We propose the diatom record preserved in ice cores from the Antarctic Peninsula and Ellsworth Land regions is the optimal proxy for reconstructing the interannual wind strength variability in the Pacific sector of the Southern Hemisphere Westerly Wind belt. Further research should be focused on expanding the study of the diatom record and its potential as a wind proxy in other regions of Antarctica and over longer timescales.

## Appendix A

**Table A1. Relative abundance (%) and frequency (# of samples) of main diatom taxa in annual diatom records for each ice core. (*) specimens of *Cyclotella* sensu lato (including *Lindavia, Discostella, Tertiarius* and *Pantocsekiella*). (s)= sea ice affiliated diatom. (o-SSIE)= open ocean – Seasonal Sea Ice Edge affiliated diatom. (o-POOZ)= open ocean – Permanently Open Ocean Zone affiliated diatom species/group. Table modified from Tetzner et al. (2021a).**

| | SHIC | JUR | SKBL |
|---|---|---|---|
| **Annual record** | **(1999-2019 CE)** | **(1992-2012 CE)** | **(1999-2019 CE)** |
| *Fragilariopsis cylindrus* (s) | **63.7 % (20)** | **18.2 % (14)** | **21.3 % (15)** |
| *Shionodiscus gracilis* (o-SSIE) | **18.5 % (18)** | **17.6 % (17)** | **10.9 % (10)** |
| *Fragilariopsis curta* (s) | **4.1 % (10)** | - | - |
| *Fragilariopsis pseudonana* (o-POOZ) | **3.6 % (11)** | **9.2 % (11)** | **15.3 % (10)** |
| *Cyclotella* group* | **6.8 % (19)** | **29.1 % (19)** | **37.2 % (20)** |
| *Navicula* group | - | **7.4 % (12)** | - |
| *Nitzschia* group | - | - | **6 % (12)** |
| *Pseudonitzschia* spp. (o-POOZ) | **3.3 % (10)** | **6.5 % (7)** | **9.3 % (8)** |
| *Achnanthes* group | - | **11.9 % (10)** | - |

**Appendix B**

Table B1. Basic statistics for the diatom abundance (n yr$^{-1}$), chemical species flux (ppb kg m$^{-2}$) and MPC flux (particles kg m$^{-2}$) annual records from each ice core. Bold numbers indicate statistically significant linear trend values ($p<0.05$). (*) MPC flux record from JUR covers the interval 1992-2011 CE due to missing data.

| | Mean | Stdv | Max | Min | Linear trend |
|---|---|---|---|---|---|
| **SHIC (1999-2019 CE)** | | | | | |
| **Diatom abundance** | 54.4 | 33.7 | 110 | 10 | 2.16 |
| **MPC flux** | 401545 | 207293 | 893535 | 130601 | 7082.97 |
| **nssCa$^{2+}$ flux** | 14898.6 | 6010.57 | 24697.7 | 6623.69 | **-622.03** |
| **nssK$^+$ flux** | 14316.9 | 10140.8 | 46777.8 | 4018.41 | -631.52 |
| **ssNa$^+$ flux** | 212145 | 74777.1 | 393636 | 73544.5 | 895.62 |
| **MSA flux** | 19383.7 | 9445.86 | 40421.8 | 1514.84 | -88.773 |
| **JUR (1992-2012 CE)** | | | | | |
| **Diatom abundance** | 57.0 | 34.0 | 166 | 20 | 2.07 |
| **MPC flux*** | 133330 | 45863.3 | 237952 | 54783.6 | 2030.22 |
| **nssCa$^{2+}$ flux** | 69827.9 | 21715.5 | 115976 | 34397.1 | **2568.96** |

| | | | | | |
|---|---|---|---|---|---|
| nssK+ flux | 14793.1 | 9887.22 | 40259.7 | 2203.81 | **1004.18** |
| ssNa+ flux | 46239.9 | 24722.7 | 110866 | 10750.2 | 1319.76 |
| MSA flux | 15543.5 | 5605.27 | 23917.6 | 5058.6 | -56.01 |
| **SKBL (1999-2019 CE)** | | | | | |
| Diatom abundance | 29.2 | 14.9 | 77 | 15 | -0.41 |
| MPC flux | 261143 | 161326 | 681947 | 52610 | **-16574** |
| nssCa2+ flux | 7662.33 | 3787.26 | 17710.2 | 2985.67 | **-315.85** |
| nssK+ flux | 3377.75 | 1317.43 | 5857 | 921.31 | **-155.21** |
| ssNa+ flux | 28285 | 17079 | 66427 | 2355.27 | -234.93 |
| MSA flux | 2927.59 | 1896.43 | 8919.92 | 773.1 | -58.28 |

**Appendix C**

Table C1. Basic statistics for the diatom abundance (n yr$^{-1}$), chemical species (ppb) and MPC (particles mL$^{-1}$) annual records from each ice core. Bold numbers indicate statistically significant linear trend values ($p<0.05$). (*) MPC record from JUR covers the interval 1992-2011 CE due to missing data.

| | Mean | Stdv | Max | Min | Linear trend |
|---|---|---|---|---|---|
| **SHIC (1999-2019 CE)** | | | | | |
| Diatom abundance | 54.4 | 33.7 | 110 | 10 | 2.16 |
| MPC | 554.80 | 287.44 | 1536.12 | 269.04 | 18.63 |
| nssCa2+ | 20.62 | 7.67 | 41.17 | 12.51 | **-0.7** |
| nssK+ | 20.56 | 14.13 | 52.65 | 6.71 | -0.48 |
| ssNa+ | 295.38 | 94.38 | 526.82 | 182.34 | 1.61 |
| MSA | 26.09 | 10.96 | 48.3 | 3.76 | -0.09 |
| **JUR (1992-2012 CE)** | | | | | |
| Diatom abundance | 57.0 | 34.0 | 166 | 20 | 2.07 |
| MPC* | 126.33 | 31.72 | 174.55 | 50.16 | 2.26 |
| nssCa2+ | 69.41 | 26.45 | 136.46 | 45.08 | **3.35** |

| | | | | | |
|---|---|---|---|---|---|
| nssK⁺ | 15.6 | 12.16 | 50.62 | 1.83 | **1.23** |
| ssNa⁺ | 43.30 | 19.84 | 85.36 | 7.64 | **2.01** |
| MSA | 15.12 | 4.56 | 24.36 | 5.58 | 0.02 |

**SKBL (1999-2019 CE)**

| | | | | | |
|---|---|---|---|---|---|
| **Diatom abundance** | 29.2 | 14.9 | 77 | 15 | -0.41 |
| **MPC** | 537.60 | 250.19 | 1239.19 | 230.46 | **-23.74** |
| **nssCa²⁺** | 15.71 | 4.43 | 24.52 | 9,48 | -0.28 |
| **nssK⁺** | 7.00 | 1.41 | 9.57 | 4.44 | **-0.2** |
| **ssNa⁺** | 61.04 | 34.94 | 137.62 | 7.44 | 1.31 |
| **MSA** | 5.91 | 2.52 | 12.35 | 2.44 | -0.01 |

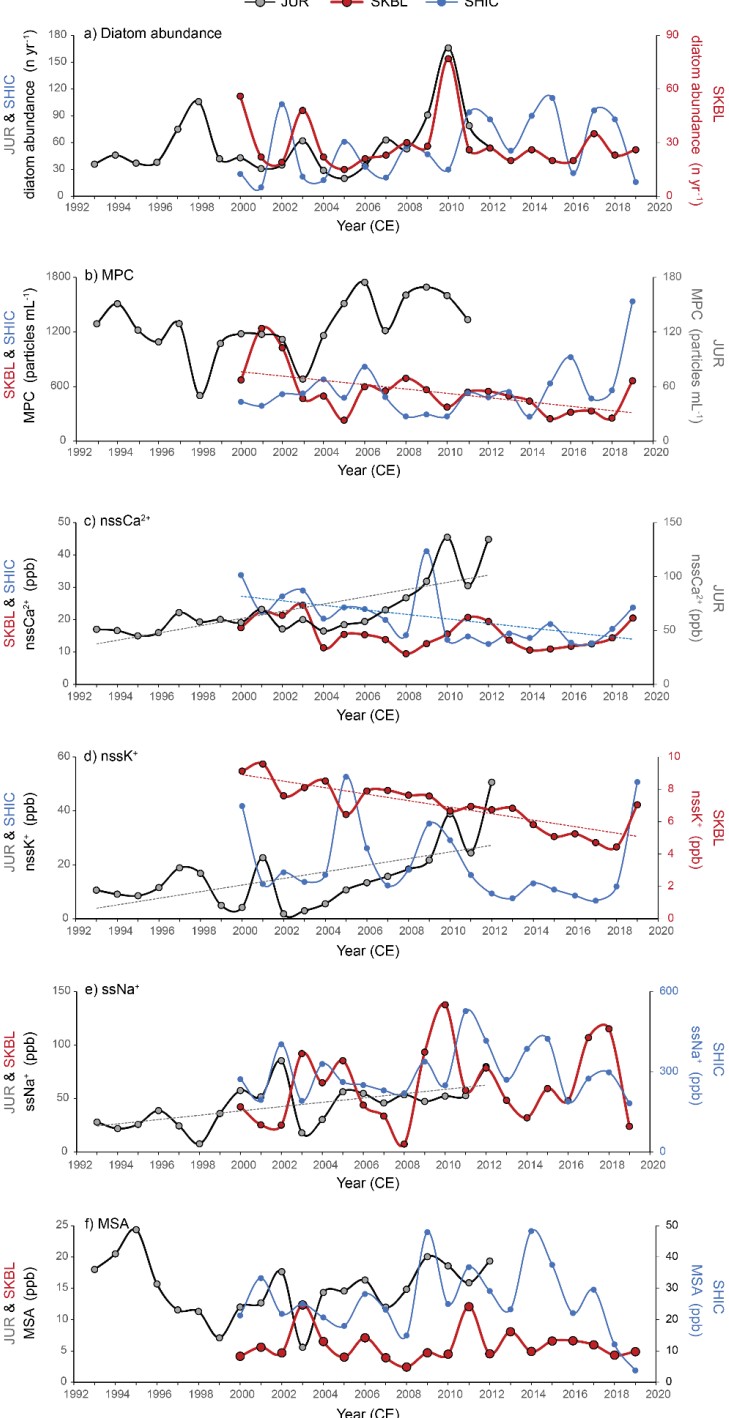

**Figure C1.** Time series of chemical species concentration, MPC and diatom abundance for each ice core site. Data points represent the annual austral winter-to-winter average and were plotted over the correspondent austral summer. Colour-coded dashed lines show statistically significant linear trends (p<0.05). MPC record from JUR covers the interval 1992-2011 CE.

## Appendix D

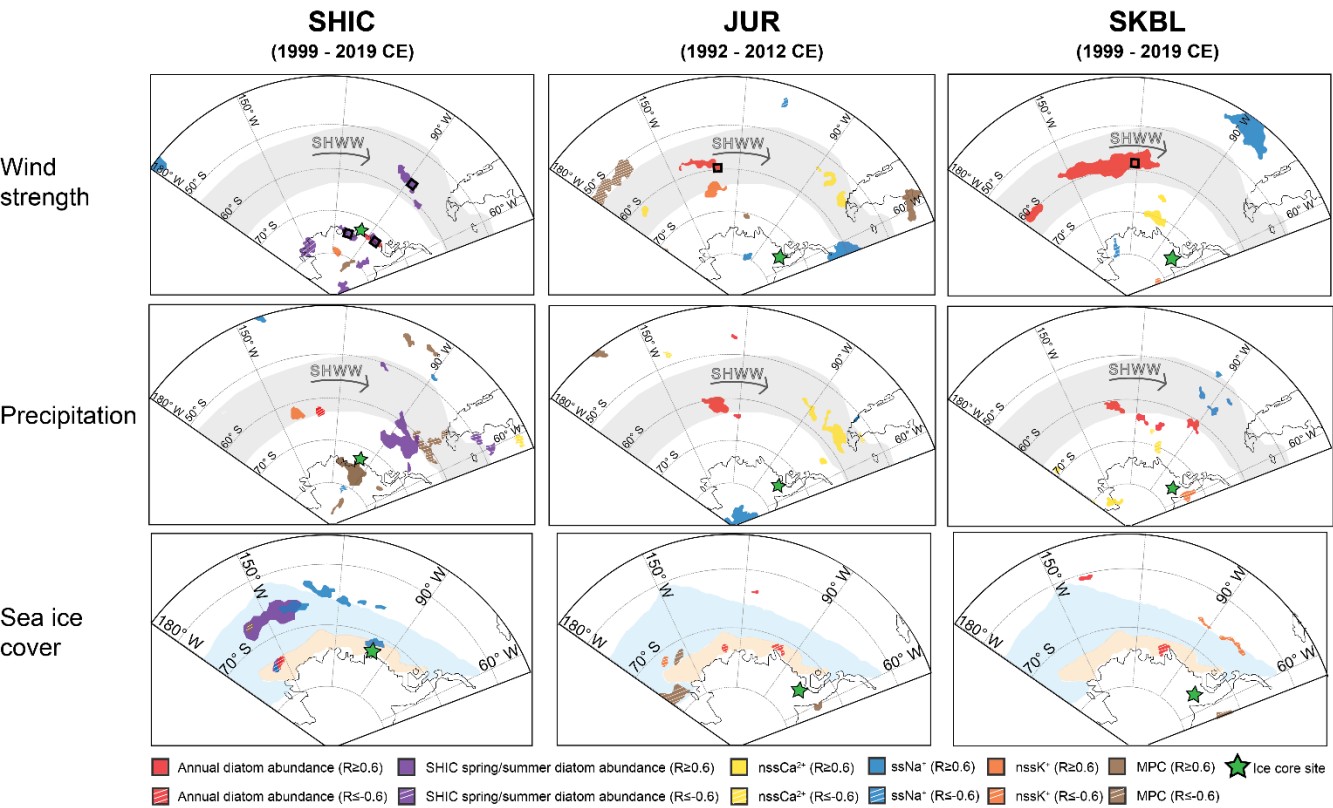

**Figure D1. Regional maps showing spatial correlations between environmental parameters and chemical species, MPC and diatom abundance. Colour-coded polygons in the maps indicate highly correlated regions (R≥0.6 or R≤-0.6). All polygons plotted are statistically significant (p<0.05). Pale grey latitudinal band indicates the core of the Southern Hemisphere Westerly Wind (SHWW) belt. Pale blue polygons indicate the region covered by seasonal sea ice. Pale orange polygons indicate the region covered by perennial sea ice. Black squares represent a 1°x1° quadrant around the point of highest correlation (QHC) between wind strength and diatom abundance. Spatial correlations between MSA and environmental parameters were excluded because they were below the threshold (R<0.6, R>-0.6 or p>0.05). Regional maps only present areas of spatial correlation larger than 1°x1°. Degrees of freedom (df) for each spatial correlation can be obtained using the following expression dependent on the sample size (n); df=n-2.**

## Data availability

Datasets original to this work will be available at the UK Polar Data Center (https://www.bas.ac.uk/data/uk-pdc/).

## Author contribution

DT did the initial conceptualization. DT and MG were in charge of data curation. DT, ET and CA conducted the formal analysis. DT was in charge of the Investigation. DT, ET and CA designed the Methodology. DT prepared the original manuscript. ET, CA and MG contributed to the reviewing and editing of the original manuscript.

**Competing interests**

The authors declare that they have no conflict of interest.

**Acknowledgements**

We would like to thank Sarah Crowsley, Tom King, Isobel Rowell and Dr Robert Mulvaney for their help while drilling the SHIC and SKBL ice cores included in this work. We would like to thank Shaun Miller, Dr Jack Humby, Dr Diana Vladimirova and Dr Daniel Emanuelsson from the Ice core Lab, British Antarctic Survey, for their help while cutting the ice and conducting
the Continuous Flow Analysis (CFA). We would like to thank Professor Eric Wolff from the Earth Sciences Department, University of Cambridge, for his valuable comments during the final review and editing of this manuscript. SEM work was partly supported by a Royal Society Research Professorships Enhancement Award (RP\EA\180006). We would like to thank Dr Iris Buisman and Dr Giulio Lampronti from the Microscopy Lab, Earth Sciences Department, University of Cambridge, for their technical support in the use of the SEM. Fieldwork conducted for this research was supported by the Collaborative
Antarctic Science Scheme (CASS). This research was funded by CONICYT–Becas Chile and Cambridge Trust funding program for PhD studies. Grant number 72180432.

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
