# Peer review of "Regional validation of the use of diatoms in ice cores from the Antarctic Peninsula as a Southern Hemisphere Westerly Wind proxy"

_Climate of the Past, 2021_

## Author Comment (AC1)

**Table 1.** Pearson's linear correlation values between the flux parameter of annual chemical species, MPC and diatom abundance for each site and across all sites. Only statistically significant values ($p < 0.05$) are included in this table. (*) MPC record from JUR covers the interval 1992-2011 CE.

| | | SHIC | | | | | | JUR | | | | | | SKBL | | | | | |
|---|---|---|---|---|---|---|---|---|---|---|---|---|---|---|---|---|---|---|---|
| | | diat | MPC | nssCa | nssK | ssNa | MSA | diat | MPC* | nssCa | nssK | ssNa | MSA | diat | MPC | nssCa | nssK | ssNa | MSA |
| SHIC | diat | - | | | | 0.81 | 0.47 | | | | | 0.59 | | | | | | | |
| | MPC | | - | 0.46 | | | | | | | | | | | | | | | |
| | nssCa | | | - | 0.48 | | | | | | | | | | | | | | |
| | nssK | | | | - | | | | 0.62 | | | | | | | | | | |
| | ssNa | | | | | - | 0.7 | | | | | | | | | | | | |
| | MSA | | | | | | - | | | | | | | | | | | | |
| JUR | diat | | | | | | | - | | 0.54 | | | | 0.69 | | | | | |
| | MPC* | | | | | | | | - | 0.45 | | 0.55 | 0.6 | | | | | | |
| | nssCa | | | | | | | | | - | 0.45 | | | | | | | | |
| | nssK | | | | | | | | | | - | | | | | | | | |
| | ssNa | | | | | | | | | | | - | 0.63 | | | | | | |
| | MSA | | | | | | | | | | | | - | | | | | | |
| SKBL | diat | | | | | | | | | | | | | - | | 0.47 | 0.46 | 0.65 | |
| | MPC | | | | | | | | | | | | | | - | 0.6 | 0.61 | | |
| | nssCa | | | | | | | | | | | | | | | - | 0.82 | | 0.75 |
| | nssK | | | | | | | | | | | | | | | | - | | 0.73 |
| | ssNa | | | | | | | | | | | | | | | | | - | 0.52 |
| | MSA | | | | | | | | | | | | | | | | | | - |

[Figure]

**Figure 1.** Map of the southern Antarctic Peninsula and Ellsworth Land highlighting the spatial distribution of ice-free areas (red surfaces) and the location of ice core sites included in this study.

---

## Author Response (AR1)

Dear Editor,

We appreciate all your comments and the comments and suggestions made by the anonymous reviewer #1 and the anonymous reviewer #2. We thank them for their time and consideration. Please find to follow a list of all points raised and our responses to each item. Reference lines mentioned in our response correspond to lines in our modified manuscript (version with "track changes" disabled).

**Anonymous reviewer 1**

**A. Reviewer 1 comment #1**

Whilst the SO is the principal source of marine diatom to this region, which is the argument allowing to rule out completely contributions from exposed sediments? Marine diatoms can be windblown to coastal sites and can be remobilized easily by winds. This does not imply that the amount of diatoms is necessarily related to the amount of dust since different dust sources can have a very different abundance of microfossils.

**Response:**

In **Tetzner et al. (2021)**, we present a detailed study of the diatom diversity found in the ice cores presented in this manuscript. The diatom records from these ice cores included marine and non-marine diatoms. These diatoms were identified to come primarily from the SO. This primary source is supported by airmass backward trajectories showing that airmasses reaching ice core sites only interact with sea level in the Northern Antarctic Zone of the SO (including the Polar front and the Permanently Open Ocean Zone) (**Thomas and Bracegirdle, 2015; Allen et al., 2020**). Despite this primary source, we cannot rule out secondary sources. Secondary sources could potentially include modern fresh/brackish-water bodies and exposed diatom-bearing sediments. We involuntarily missed including in the original manuscript details about the regional diatom diversity and mention potential sources (other than the primary marine source). We agree with the reviewer that we cannot rule out contributions either from exposed sediments or from modern non-marine waterbodies.

To address this comment, we have modified the manuscript removing references to diatoms as exclusively marine (**e.g. Line 62, Line 76**). Additionally, we have included further details about the diatom diversity of these ice cores (**Lines 130-133 and Appendix A-Table A1**) and outlined potential sources of diatoms in the Antarctic region (**Lines 67-68**).

**References**

Allen, C. S., Thomas, E. R., Blagbrough, H., Tetzner, D. R., Warren, R. A., Ludlow, E. C., and Bracegirdle, T. J.: Preliminary Evidence for the Role Played by South Westerly Wind Strength on the Marine Diatom Content of an Antarctic Peninsula Ice Core (1980–2010), Geosciences, 10, 87, https://doi.org/10.3390/geosciences10030087, 2020

Tetzner, D. R., Thomas, E. R., and Allen, C. S.: Marine diatoms in ice cores from the Antarctic Peninsula and Ellsworth Land, Antarctica; species diversity and regional variability, The Cryosphere Discussions, 1–32, https://doi.org/10.5194/tc-2021-160, 2021.

Thomas, E. R. and Bracegirdle, T. J.: Precipitation pathways for five new ice core sites in Ellsworth Land, West Antarctica, Clim Dyn, 44, 2067–2078, https://doi.org/10.1007/s00382-014-2213-6, 2015.
* * *
**A. Reviewer 1 comment #2**

Why the abundance of freshwater or brackish-water diatoms is not taken into account? It seems completely neglected but it can be very useful. Are sponge spicules or other microfossils also present in the samples? Is it possible to show the two diatom abundance records (marine and nonmarine/brackish)? Also, which species of diatoms are you looking at? Imagine that not all readers switch between the two papers in order to understand what you are effectively counting.

**Response:**

The diatom abundance presented in this manuscript includes all marine and fresh/brackish-water diatoms found on each ice core. We agree the non-marine component of the assemblage could hold valuable information. However, as reported in the review of **Tetzner et al. (2021)** (https://doi.org/10.5194/tc-2021-160-AC1), insufficient image resolution prevented us from unequivocally differentiating between marine and non-marine species in four "cosmopolitan" genus. These limitations prevented us from accurately quantify the non-marine proportion of diatoms on each ice core site. Despite this, if it was assumed that all the diatoms classified in the cosmopolitan groups were non-marine, the main diatom assemblage at each ice core site would still be prevalently conformed by marine diatoms (>58%). Sponge spicules were not identified in our samples. Our samples occasionally presented low numbers of chrysophyte stomatocytes.

To address this comment, we have included, in the method section, a sentence specifying the diatom abundance accounts for all marine, non-marine diatoms and indistinctive diatom fragments found on each ice core (**Lines 127-130**). We have added a table (**Appendix A – Table A1**) detailing which species are present on each ice core and their correspondent proportion of the main diatom assemblage, as reported in **Tetzner et al. (2021)**. We have also modified the manuscript removing all references to marine diatoms and replacing them for "diatoms" (**e.g. Line 62, Line 76**).

**References**

Tetzner, D. R., Thomas, E. R., and Allen, C. S.: Marine diatoms in ice cores from the Antarctic Peninsula and Ellsworth Land, Antarctica; species diversity and regional variability, The Cryosphere Discussions, 1–32, https://doi.org/10.5194/tc-2021-160, 2021.
* * *
**A. Reviewer 1 comment #3**

Are the marine species identified in the cores comparable to marine species which are

found in typical Sirius formation?

**Response:**

The main diatom assemblage identified in the Antarctic Peninsula (AP) and Ellsworth Land (EL) ice cores was comprised of *Fragilariopsis cylindrus, Shionodiscus gracilis, Fragilariopsis curta, Fragilariopsis pseudonana, Cyclotella* gp*., Navicula* gp*., Nitzschia* gp*., Pseudonitzschia* spp., and *Achnanthes* gp. (**Information now included in Appendix A – Table A1**). Of them, only *Nitzschia* spp. was identified in the Sirius formation (**Harwood, 1983; Harwood, 1986; McKay et al., 2008**). Diatoms classified as *Nitzschia* spp. are scarcely present in the AP and EL ice cores. They were only identified in the SKBL ice core and accounted for 6% of the main diatom assemblage at that site (**Tetzner et al., 2021**) (**Information now included in Appendix A – Table A1**). The lack of common diatoms evidence a weak relationship between the marine species found in the Sirius group and the ones found in the AP and EL ice cores. In turn, the main diatom assemblage in ice cores from the AP and EL region closely resembles the main diatom assemblage reported by **Budgeon et al. (2012)** in fresh snow samples obtained near Casey Station, which included *F. cylindrus, S. gracilis, F. curta, F. pseudonana, Cyclotella* gp*., Navicula* gp*., Nitzschia* gp*.

To address this comment, we have added a Table (**Appendix A - Table A1**) outlining the main diatom assemblage at each ice core site.

**References**

Budgeon, A. L., Roberts, D., Gasparon, M., and Adams, N.: Direct evidence of aeolian deposition of marine diatoms to an ice sheet, Antarctic Science, 24, 527–535, https://doi.org/10.1017/S0954102012000235, 2012

Harwood, D. M. (1983). Diatoms from the Sirius Formation, Transantarctic Mountains. Antarctic Journal of the United States, 18(5), 98-100.

Harwood, D. M. (1986). Recycled siliceous microfossils from the Sirius Formation. Antarctic Journal of the United States, 21(5), 101-103.

McKay, R. M., Barrett, P. J., Harper, M. A., & Hannah, M. J. (2008). Atmospheric transport and concentration of diatoms in surficial and glacial sediments of the Allan Hills, Transantarctic Mountains. Palaeogeography, Palaeoclimatology, Palaeoecology, 260(1-2), 168-183.

Tetzner, D. R., Thomas, E. R., and Allen, C. S.: Marine diatoms in ice cores from the Antarctic Peninsula and Ellsworth Land, Antarctica; species diversity and regional variability, The Cryosphere Discussions, 1–32, https://doi.org/10.5194/tc-2021-160, 2021.

**A. Reviewer 1 comment #4**

In the introductions, lines 66-67, you cite as references for the sentence "Once in the atmosphere, they can be transported by winds over long distances" the papers from Gayley, 1989 and McKay et al., 2008 [I have no access to the 3rd publication cited]. Both papers consider diatoms as associated mainly with deflation of dry sediments. So, please provide references clearly indicating sea spray as the primary source for long-range transported marine diatoms.

**Response:**

Revised as suggested. We modified the previous references and added new ones that support sea sprays as the source for long-range transported marine diatoms (**Lines 70-71**).
* * *
**A. Reviewer 1 comment #5**

In your statistical analysis, The diatom abundance ($n*a_{-1}$) is a sort of annual depositional flux of diatoms (number of specimens per year) that already takes into account the snow accumulation rate, since it is calculated over an entire year. For the chemical parameters, conversely, you use average concentrations per year (I see "ppb" in your figure!??), not fluxes..(?). So if concentrations are used instead of depositional fluxes, how can you get free from the snow accumulation rate for chemicals? All data must be transformed into fluxes otherwise the comparison of chemical/dust records among different sites has no sense.

**Response:**

We agree with the reviewers comment. To address this comment, we have modified the results section to report the chemical and MPC fluxes (along with diatom abundance) (**Lines 168-177, 204-209, 221-228 and 243-250**). We modified Figure 2 and Table B1, which now present diatom abundance, and the chemical and MPC fluxes **(Line 178 and Appendix B)**. Additionally, we modified Table S1, which now presents the correlation of the flux parameter of ice core records (**Supplementary material – Table S1**).

Fluxes were used when comparing records from the three ice core sites. Conversely, spatial correlations obtained when comparing ice core records with environmental parameters were calculated using chemical concentrations and MPC (these concentration records are now presented in **Appendix C - Table C1, and Appendix C - Figure C1**). This approach was followed to prevent incorporating the variability from secondary parameters (snow accumulation) to spatial correlations. Our approach is supported by the results obtained from the recent "CLIVASH2k-ice core chemistry" initiative, which gathered $Na^+$ and $SO_4^{2-}$ data (concentrations and fluxes) from over 100 ice core sites across Antarctica (**Thomas et al., in prep**). Data collected for this initiative show that the flux parameter in the Antarctic Peninsula leads to comparatively weaker and inconsistent spatial correlations with environmental parameters. Thus, suggesting the flux parameter in the Antarctic Peninsula region is biased by the high snow accumulation interannual variability.
* * *
**A. Reviewer 1 comment #6**

When looking at figure 2, one can observe that JUR and SKBL nicely show similar diatom abundance variability that is quite obvious given the location of the two sites and their common sensitivity to open ocean species. The SHIC core instead shows a different pattern of variability since it is sensitive to sea ice taxa. These are conclusions from the companion paper Tetzner 2021a. So, I think it is not well clear in this work what is novel and what is part of the conclusions drawn in the companion paper.

**Response:**

Figure 2 presents this data to show the reader the interannual variability of the diatom abundance at each ice core site. Despite this data was presented in **Tetzner et al. (2021)**, we decided to include it in Figure 2 for the reader to link the interannual variability of the diatom abundance to the values presented in **Appendix B - Table B1, Appendix C - Table C1** and **Supplementary material - Table S1** (calculations which were not presented in **Tetzner et al. 2021**).

To address this comment, we have specified in the method section that diatom abundance records were previously presented in **Tetzner et al. (2021)** (**Line 122**) and we specified in Figure 2 caption that the data presented in panel (a) was already presented in **Tetzner et al. (2021)** (**Lines 181-182**).

**References**

Tetzner, D. R., Thomas, E. R., and Allen, C. S.: Marine diatoms in ice cores from the Antarctic Peninsula and Ellsworth Land, Antarctica; species diversity and regional variability, The Cryosphere Discussions, 1–32, https://doi.org/10.5194/tc-2021-160, 2021.
* * *
**A. Reviewer 1 comment #7**

The attribution of JUR and SKBL marine diatoms to the POOZ suggested by Tetzner (TC, 2021a) is interesting, but given the very coarse size of such marine diatoms, a mechanism for strong uplift and transport inland is required. So, is it possible that diatom abundance reflects not only wind strength *sensu stricto* but low-pressure systems generated in that POOZ area or passing through that area and then directed towards the Peninsula? Indeed wind strength around LP systems is generally higher, so it is just a different interpretation of this correlation.

**Response:**

The mechanism proposed by the reviewer is plausible and has been previously considered by the authors, but not mentioned in this manuscript. We did not propose this mechanism for uplift and transport because we did not present data that could directly support it (e.g. spatial correlation between diatom abundance and mean sea level pressure and/or 850 hPa height). We did not include a comparison between diatom abundance and mean sea level pressure because we wanted to focus our discussion in the relationship between wind strength and diatom abundance. We agree with the reviewer that our manuscript does not mention explicitly a mechanism for the uplift and transport of diatoms to inland sites. However, as mentioned by the reviewer, it is implicit that strong winds are intrinsically linked to low-pressure systems which actively uplift and delineate airmass transport pathways in this region.
* * *
**A. Reviewer 1 comment #8**

The novel proxy for wind strength is interesting but different from proxies like Calcium and dust. It is not correct to say (330-333) that particles and calcium reflect wind strength as they have been always associated with the cumulative effect of different factors that are: the primary production at the source(s), the humidity/precipitation en route during atmospheric transport, the snow accumulation rate in Antarctica, ... Conversely, a proxy that is much more directly related to transport (including wind

strength) is dust grain size. So it is not correct to say that these are traditional proxies for wind strength. Different proxies are related to different dynamics. Please change these considerations accordingly.

**Response:**

We agree with this comment. To address this comment, we have modified the manuscript accordingly. We now emphasise that major ions and dust have been traditionally used to interpret changes in atmospheric circulation in a broad sense, not restricted to wind strength (**e.g. Lines 74-75, Line 79**).
* * *
**A. Reviewer 1 comment #9**

In general, it must be clarified to the reader that given the position of the sites, dust and Calcium are probably dominated by the effect of the local dust sources from marginal ice-free areas, that are not the same sources of marine diatoms but can provide diatoms through eolian reworking.

**Response:**

The main source of insoluble dust, calcium ($nssCa^{2+}$) and potassium ($nssK^+$) to this region has not been yet well established in the literature. It has been suggested that the main contributors of dust and calcium to this region could be (1) southern South America (SA) (mainly from the Patagonia region), (2) New Zealand/Australia, and (3) local Antarctic sources (**McConnell et al., 2007; Bory et al., 2010; Koffman and Kreutz, 2014; Neff and Bertler, 2015; Bullard et al., 2016**). The largest source of dust in Antarctica are the Transantarctic Mountains and the McMurdo Dry Valleys region (TAMS-MDV) (**Bullard et al., 2016**). Both, SA and the TAMS-MDV are located 1500-2000 km away from the ice core sites, highlighting both as potential contributors of dust and calcium. Small ice-free areas are scattered across the Antarctic Peninsula (AP). The sum of all these areas accounts for less than 3% of the total surface of the AP (**Siegert et al., 2019**). A small number of ice-free areas are located within a 100-km radius from the ice core sites (**see Figure 1 attached**). However, these areas are very small and not exposed to active weathering processes, preventing them from contributing considerable amounts of dust to the ice core sites.

Among the three ice core sites, SKBL is the more proximal to ice-free areas. Despite its proximity to ice-free areas, SKBL does not exhibit a considerably larger amount of dust compared to SHIC, which lacks of ice-free areas on its vicinities. Similar dust values at both sites suggest ice-free areas near JUR and SKBL do not play a major role supplying dust to the ice core sites. In light of this, we do not share this reviewers comment: "dust and calcium are probably dominated by the effect of the local dust sources from marginal ice-free areas". We cannot rule out potential contributions of dust from local ice-free areas to ice core sites. However, we do not support local ice-free areas are the primary source of dust to these sites. Instead, we support a major proportion of dust and calcium to be originated from distal sources (>1000 km) with secondary contributions from neighbouring ice free areas, in line with the results obtained by **McConnell et al. (2007)** in the northern Antarctic Peninsula.

To address this comment, we have incorporated new information in the Introduction section to emphasize that dust and calcium can be originated from both local and distal sources (**Lines 48-50**). We have also outlined the various sources from where diatoms could have been removed to then become part of the ice core record (**Lines 67-68**). This new information was included for the reader to know that there are other, secondary, sources of diatoms which could potentially contribute to shape the diatom abundance parameter.

[Figure]

**Figure 1.** Map of the southern Antarctic Peninsula and Ellsworth Land highlighting the spatial distribution of ice-free areas (red surfaces) and the location of ice core sites included in this study.

**References**

Bory, A., Wolff, E., Mulvaney, R., Jagoutz, E., Wegner, A., Ruth, U., & Elderfield, H. (2010). Multiple sources supply eolian mineral dust to the Atlantic sector of coastal Antarctica: Evidence from recent snow layers at the top of Berkner Island ice sheet. Earth and Planetary Science Letters, 291(1-4), 138-148.

Bullard, J. E., Baddock, M., Bradwell, T., Crusius, J., Darlington, E., Gaiero, D., ... & Thorsteinsson, T. (2016). High-latitude dust in the Earth system. Reviews of Geophysics, 54(2), 447-485.
Koffman, B. G., & Kreutz, K. J. (2014). Evidence that local dust sources supply low-elevation Antarctic regions. Past Global Changes Magazine, 22(2), 76-77.

Koffman, B.G. and Kreutz, K.J., 2014. Evidence that local dust sources supply low-elevation Antarctic regions. Past Global Changes, 22(2), 76-77

McConnell, J. R., Aristarain, A. J., Banta, J. R., Edwards, P. R., & Simões, J. C. (2007). 20th-Century doubling in dust archived in an Antarctic Peninsula ice core parallels climate change and desertification in South America. Proceedings of the National Academy of Sciences, 104(14), 5743-5748.

Neff, P. D., & Bertler, N. A. (2015). Trajectory modeling of modern dust transport to the Southern Ocean and Antarctica. Journal of Geophysical Research: Atmospheres, 120(18), 9303-9322.

Siegert, M. J., Kingslake, J., Ross, N., Whitehouse, P. L., Woodward, J., Jamieson, S. S., ... & Sugden, D. E. (2019). Major ice sheet change in the Weddell Sea sector of West Antarctica over the last 5,000 years. Reviews of Geophysics, 57(4), 1197-1223.
* * *
**A. Reviewer 1 comment #10**

Since dust deposited at JUR likely comes from proximal sources (and to a lesser extent from remote areas) I cannot find sense in the correlation between dust at JUR and wind strength 10m altitude around 40-45°S. Also, dust from remote continents must travel at high elevations in order to reach Antarctica. So, again I am not sure that all correlations that are shown in figure 3 make sense and are worth to be considered.

**Response:**

As previously stated (see response to A. reviewer 1 comment #9):

"We do not support local ice-free areas are the primary source of dust to these sites. Instead, we support a major proportion of dust and calcium to be originated from distal sources with secondary contributions from neighbouring ice free areas"

Our statement is further supported by numerous previously published work. In particular, **Li et al. (2010)** demonstrated dust can be transported within ~4-5 days, in the low/mid troposphere, from South America to the Antarctic Peninsula and West Antarctica. Similarly, **Koffman et al. (2017)** demonstrated coarse ash from a volcanic eruption that occurred in South America (40°S) was effectively transported in the low/mid troposphere to the WAIS Divide ice core site (79.5°S) in West Antarctica within 7 days after the initial eruption. Additionally, several studies support airmasses from South American can take 5-10 days to reach the Antarctic Peninsula and Ellsworth Land (**Abram et al., 2010; Neff and Bertler, 2015; Thomas and Bracegirdle, 2015).** Altogether, these lines of evidence support the correlation between JUR dust and wind speed at 40-45°S is plausible.

**References**

Abram, N. J., Thomas, E. R., McConnell, J. R., Mulvaney, R., Bracegirdle, T. J., Sime, L. C., and Aristarain, A. J.: Ice core evidence for a 20th century decline of sea ice in the Bellingshausen Sea, Antarctica, Journal of Geophysical Research: Atmospheres, 115, https://doi.org/10.1029/2010JD014644, 2010.

Koffman, B. G., Dowd, E. G., Osterberg, E. C., Ferris, D. G., Hartman, L. H., Wheatley, S. D., ... & Yates, M. (2017). Rapid transport of ash and sulfate from the 2011 Puyehue-Cordón Caulle (Chile) eruption to West Antarctica. Journal of Geophysical Research: Atmospheres, 122(16), 8908-8920.

Li, F., Ginoux, P., & Ramaswamy, V. (2010). Transport of Patagonian dust to Antarctica. Journal of Geophysical Research: Atmospheres, 115(D18).

Neff, P. D., & Bertler, N. A. (2015). Trajectory modeling of modern dust transport to the Southern Ocean and Antarctica. Journal of Geophysical Research: Atmospheres, 120(18), 9303-9322.

Thomas, E. R. and Bracegirdle, T. J.: Precipitation pathways for five new ice core sites in Ellsworth Land, West Antarctica, Clim Dyn, 44, 2067–2078, https://doi.org/10.1007/s00382-014-2213-6, 2015.
* * *
**A. Reviewer 1 comment #11**

Line 78 - Are these really ice cores or firn cores?

**Response:**

SHIC and SKBL are firn cores. The top section of JUR included in this study also corresponds to firn (reaching a density of 700 kg m$^{-3}$ at 36.9 meters deep). We modified the manuscript specifying this as suggested (**Lines 82-86**). To highlight the wide scope of our results we decided to treat firn cores and ice cores indistinctively. For practicalities, we have added a caveat specifying that the manuscript will use the term "ice cores" when referring to "firn cores" (**Line 86**).
* * *
**A. Reviewer 1 comment #12**

Line 115: if microparticles are measured with an Abakus sensor, it is possible to get an idea of the degree of sorting of the dust, that is useful to constrain sources and transport distance?

**Response:**

We agree a detailed analysis of the particle size distribution and the variability of the finer/coarser fraction of dust can contribute to constrain sources and transport distances for dust. However, the aim of the research work presented here is to evaluate the potential for diatoms to reconstruct regional wind strength. The incorporation of other traditional wind and atmospheric circulation proxies was to validate the novel diatom proxy and to compare the performance of these proxies in the AP-EL regions. Based on the aims of this study, the incorporation of a detail study of the PSD and size subsets is beyond the scope of this work.
* * *
**A. Reviewer 1 comment #13**

Line 120: Can small diatom fragments (that you discard from your counts) provide an idea of the degree of diatom reworking?

**Response:**

Diatom fragments discarded from our counts correspond to every fragment smaller than 5 microns in its longest axis. Unequivocally differentiating diatom fragments below 5 microns from other insoluble particles is already a major challenge, regardless of their degree of reworking. The difficulties of identifying diatom fragments of these sizes arise from the incapacity of diatom fragments of these sizes to retain features that will allow to identify their diatom origin. Thus, the discarded fraction will not provide a conclusive idea of the degree of diatom reworking.

**A. Reviewer 1 comment #14**

Line 121: "Diatom abundance" means marine-only diatoms or really "all diatom valves"?

**Response:**

Please remit to our response to "reviewers comment #2"

**A. Reviewer 1 comment #15**

Paragraph 3.1.2: The correlation between diatom abundance per year and wind strength is interesting and is probably one of the key new messages of this work. However, figure 3 is too rich and the attention of the reader is not immediately captured by that. I also wonder if many of these correlations make sense. I suggest splitting this figure in order to focus on the most interesting part of it while moving the remaining part to the supplementary information.
For example, both JUR and SKBL show a correlation between diatom flux and wind strength, while the correlations related to Calcium and dust that are found at one site are very different from the other, and in any case, they are difficult to understand. Is there a possible bias related to the use of average concentrations instead of depositional fluxes? Actually, in line 227 you mention that "No clear or consistent pattern was identified when comparing chemical proxies from different ice core sites" – and this is quite strange when JUR and SKBL are considered.

**Response:**

To address this comment, we followed the reviewers suggestion and modified **Figure 3** to focus on the spatial correlations between diatom abundance and environmental parameters, while moving the initially submitted figure to Appendix D.

**Anonymous reviewer 2**

**A. Reviewer 2 comment #1**

**Section #1.1**

It is mentioned lines 330-335 that established ice core wind proxies such as nssCa, ssNa and nssK present very patchy spatial correlations with annual wind strength across the southern mid-latitudes (figure 3). For this reason, these proxies are mentioned to be of limited interest in this region. However, it appears to be exactly the same for diatom abundances who similarly show very patchy spatial correlations with wind strength (SHIC) and very restricted zones of high correlation (JUR, especially, and SKBL, figure 3).

**Response #1.1:**

Our results evidence traditional ice core wind and atmospheric circulation proxies present limitations to reproduce the interannual variability of wind strength in the Pacific core of the SHWWs. In particular, these traditional proxies exhibited patchy regions of high-correlation (R>0.6 or R<-0.6, p<0.05) with wind strength outside the core of the SHWWs. Additionally, the locations of most of these regions of high correlation did not fit with their expected sources and were not consistent between sites. Based on the lack of consistency and correlation between each of these proxies and wind strength within the SHWWs, we discard them as potential indicators of wind strength variability in the SHWW. Altogether, our results highlighting these traditional ice core proxies represent the cumulative effect of numerous factors related to atmospheric transport and source conditions, not strictly wind strength in the core of the SHWWs. In parallel, the diatom abundance from high elevation sites in the AP shows a regionally consistent area of high correlation in the core of the SHWWs (Ferrigno ice core in **Allen et al. (2020)** and JUR and SKBL in Figure 3).

The reviewer states that SKBL and JUR exhibit "very restricted zones of high correlation". The restricted appearance of these regions arises from the fact that in the originally submitted Figure 3, we only plotted the regions that exhibited a correlation R>0.6 or R<-0.6. The reason why we only plotted areas of spatial correlation that exhibited values R>0.6 and R<-0.6 was to make Figure 3 less clumped. However, we acknowledge that by presenting the data this way, we have possibly undermined the information we wanted to present.

To address this comment, and following a previous comment from Anonymous Reviewer #1 (comment #15), we have modified Figure 3 and included the whole area of statistically significant correlations (p<0.05) for diatom abundance (**see new Figure 3**). We have moved the original Figure 3 to **Appendix D**, where the reader will be able to identify all the areas that exhibited high correlations (R>0.6 or R<-0.6), regardless of their consistency. Additionally, we have modified the manuscript to specify that the limitations we report from traditional ice core wind and atmospheric circulation proxies are mainly on their capacity to represent wind strength variability within the SHWW.

**References**

Allen, C. S., Thomas, E. R., Blagbrough, H., Tetzner, D. R., Warren, R. A., Ludlow, E. C., & Bracegirdle, T. J. (2020). Preliminary evidence for the role played by South Westerly wind strength on the marine diatom content of an Antarctic Peninsula Ice Core (1980–2010). Geosciences, 10(3), 87.

**Section #1.2**

Additionally, for JUR and SKBL, these quadrants of high correlations (QHCs) are outside the main 950 hPa circulation to the ice cores as evidence in Allen et al. (2020). Indeed, back trajectories showed very low density in the 50-60°S band west of 120°W (figure 6, Allen et al., 2020). So one may question how diatom can be sea-sprayed from the NAZ between 120-150°W and then transported to the ice core sites if the air masses reaching the core sites do not sweep the QHCs? Somehow, the correlation between the diatom abundance records and the QHCs might not be causal. One may imagine that changes in wind strength in the QHCs (zonal circulation) also increases the strength of the meridional circulation, which allows a greater diatom deposition in the coastal ice cores. However, nothing proves that there is a transfer of diatoms from the zonal circulation to the meridional circulation (see after with the comment on the diatom assemblages preserved in the ice cores).

**Response #1.2:**

The regions of high correlation (R>0.6) between diatom abundance and wind strength (for JUR and SKBL) are mainly constrained within 55-60°S and 120-150°W. Similarly, the region of high correlation (R>0.6) presented by **Allen et al. (2020)** for the Ferrigno ice core is constrained within 55-60°S and 120-140°W. These quadrants of high correlation lie within the region where 20-40% of the air masses reaching the Ferrigno site potentially entrain marine aerosols **(Allen et al., 2020)**. The reviewer questions how diatoms from the identified region of high correlations can be transported to the ice core sites if the percentage of air masses interacting with this region is so "low" (20-40%).

The reviewer's questionings are based on the 5-day back-trajectory analysis performed by **Allen et al. (2020)** for the Ferrigno ice core site, a site that represents an analogue of the high elevation sites presented in this manuscript (**Thomas and Bracegirdle, 2015**). In particular, the questioned percentages represent the spatial distribution of air masses reaching high elevation ice core sites. In light of these questionings, it must be highlighted that the percentages presented will inevitably decrease the further away a point is from the final destination (ice core site), as trajectories will be dispersed over a larger area. The closer trajectories are from the final destination, the more channelled they are, increasing the percentages. A percentual reduction with increasing distance indicates that fewer air masses are passing through a unit of surface, not necessarily implying that a considerable number of trajectories are not coming from a large distal region (e.g. our region of high correlation). Another point worth noting is that the back-trajectory analyses presented by **Allen et al. (2020)** are based on 5-day trajectories. Since airmasses do not necessarily move in straight lines, increasing the number of days considered in the back-trajectory analysis would likely increase the percentages of airmasses travelling through distal areas.

The evidence of airmasses passing through the high correlation region enables us to establish causality between wind strength at the QHCs and diatom abundance at the ice core sites. In particular, it has been widely demonstrated that strong winds over the surface of the ocean enhance the production of sea-spray aerosols (including diatoms (**Marks et al., 2019**)) (**Andreas, 1992; Wu, 1993; Andreas et**

**al., 1995; Anguelova et al., 1999; O'Dowd and De Leeuw, 2007**). The enhanced production of sea-sprays is regardless of wind direction. The considerable number of trajectories passing over the high correlation region and their subsequent transport south, following regional atmospheric circulation patterns (**Allen et al., 2020**), establishes a mechanism to link stronger winds (enhanced sea-spray production) with the higher abundance of diatoms in the ice core sites. Thus, supporting a cause-effect relation.

**References**

Allen, C. S., Thomas, E. R., Blagbrough, H., Tetzner, D. R., Warren, R. A., Ludlow, E. C., & Bracegirdle, T. J. (2020). Preliminary evidence for the role played by South Westerly wind strength on the marine diatom content of an Antarctic Peninsula Ice Core (1980–2010). Geosciences, 10(3), 87.

Andreas, E. L. (1992). Sea spray and the turbulent air-sea heat fluxes. Journal of Geophysical Research: Oceans, 97(C7), 11429-11441.

Andreas, E. L., Edson, J. B., Monahan, E. C., Rouault, M. P., & Smith, S. D. (1995). The spray contribution to net evaporation from the sea: A review of recent progress. Boundary-Layer Meteorology, 72(1), 3-52.

Anguelova, M., Barber Jr, R. P., & Wu, J. (1999). Spume drops produced by the wind tearing of wave crests. Journal of Physical Oceanography, 29(6), 1156-1165.

Marks, R., Górecka, E., McCartney, K., & Borkowski, W. (2019). Rising bubbles as mechanism for scavenging and aerosolization of diatoms. Journal of Aerosol Science, 128, 79-88.

O'Dowd, C. D., & De Leeuw, G. (2007). Marine aerosol production: a review of the current knowledge. Philosophical Transactions of the Royal Society A: Mathematical, Physical and Engineering Sciences, 365(1856), 1753-1774.

Thomas, E. R., & Bracegirdle, T. J. (2015). Precipitation pathways for five new ice core sites in Ellsworth Land, West Antarctica. Climate dynamics, 44(7-8), 2067-2078.

Wu, J. (1993). Production of spume drops by the wind tearing of wave crests: The search for quantification. Journal of Geophysical Research: Oceans, 98(C10), 18221-18227.

**A. Reviewer 2 comment #2**

The statistics are based on a small number of data in each ice core. At SKBL, which shows the strongest and largest spatial correlation between diatom abundance and wind strength, it is based on 19 samples. I wonder whether this is statistically significant, especially as the reader is not told how the degrees of freedom for the tests were determined, and what allowance was made for the autocorrelation in the relevant series. This statistical aspect should be presented much more rigorously. See, e.g., Bretherton et al., 1999, The effective number of spatial degrees of freedom of a time-varying field. Journal of Climate, 12, 1990-2009. In the same vein, I did not really get how the series were detrended. Only by subtracting the first order linear trend? But many records do not show

any trend. And sometimes mentioned trends are not evident. For example, one really needs the eye of the believer to see any trend in diatom abundance, nssCa, ssNa and MSA in core SKBL, despite what is written line 184.

**Response:**

All spatial correlations presented in this manuscript are based on 20-year periods (20 data points per site (**Lines 134-135 and Lines 160-161**)), unless the opposite was stated (e.g. **Lines 161-162** state that JUR MPC record is comprised by 19 data points). Every value reported as statistically significant (p<0.05) passed both, one-tail and two-tailed tests. Spatial correlations were calculated using the field correlation tool from the Royal Netherlands Meteorological Institute - KNMI Climate Explorer.

All datasets presented in this manuscript were detrended by subtracting the first order linear trend before calculating correlations. This procedure was conducted regardless of the dataset trends being statistically significant (p<0.05). The linear detrending was performed to remove potential trends which could bias the calculation of correlation coefficients. Linear trends are included in the results section to present the datasets that were used for calculating correlations. The original manuscript specifies which trends are statistically significant (p<0.05).

No specific allowances were made for the autocorrelation of the datasets. All the raw (sub-annual) MPC, chemical and diatom records obtained from ice cores and wind speed and precipitation reanalyses products were not autocorrelated (-0.3<R<0.3) over the time intervals analysed. The raw data of the sea ice cover parameter from reanalysis products exhibited an autocorrelation, possibly due to its strong seasonal cycle.

To address this comment we have modified the manuscript specifying the degrees of freedom for each set of correlations calculated (**Lines 200-201, and caption in Supplementary material – Table S1**). We also included, in the methods section, details about the algorithm we used to obtain the spatial correlations (KNMI Climate explorer) (**Lines 154-156**). Finally, we specified in the methods section that all datasets were detrended by subtracting the first order linear trend (**Line 134 and Line 156**).
* * *
**A. Reviewer 2 comment #3**

The diatom assemblages appear as important to deal with as the total diatom abundances. They are presented in Tetzner et al. (2021a), which commits the reader to uneasily shuffle between the two manuscripts. It is mentioned lines 237-239 that the QHC regions match with the production zones of the main diatom species preserved in the ice cores. I somehow disagree with that general statement. More specifically, in JUR only S. gracilis (30% of the total diatom assemblages) is produced in the open ocean NAZ. Fragilariopsis cyclindrus is produced within the seasonal sea ice zone, south of 60°S. Fragilariopsis pseudonana occurs in high abundances around the South Shetland Islands. The Cyclotella, Achanthes and Navicula groups (> 50% of the diatom assemblages) represent diatom thriving at the AAP and AS-BS coasts, maybe be at the South American coast. The same interpretations can be drawn for SKBL. This fits quite well with the back-trajectories presented in Allen et al. (2020) with highest density along the 80°W parallel. This suggests that wind strength might not be the only (main?) driver of diatom transport and deposition in coastal ice cores. The wind direction is also very (most?) important.

**Response:**

**Tetzner et al. (2021)** present the main diatom assemblages identified for each ice core site included in this manuscript. The main diatom assemblage identified in JUR and SKBL is composed of two groups of diatoms. These include (1) a group conformed by exclusively marine diatom species, and (2) a group of diatoms that were only possible to identify to Genus level, therefore, not allowing to differentiate between marine (open ocean) and freshwater/brackish species. Among the exclusively marine group (1), there were sea ice affiliated diatoms and open ocean affiliated diatoms (species found within and south of the Antarctic Polar Front). Since the proportion of marine diatoms in (2) is unknown, the proportion of identified marine diatoms (1) is the following: Open ocean diatoms within the NAZ account for at least 65% and 63% of the marine diatoms identified in JUR and SKBL, respectively. In turn, sea ice diatoms account for at the most 35% and 37% of the marine diatoms identified in JUR and SKBL, respectively. These proportions evidence open ocean diatoms within the NAZ account for the majority of marine diatoms present in the high-elevation ice cores. This evidence supports the QHC (55-60°S, 120-150°W) match with the production zone of the main marine diatom species.

To address this comment, we have added a table (**Appendix A – Table A1**) detailing which species are present on each ice core and their correspondent proportion of the main diatom assemblage, as reported in **Tetzner et al. (2021)**. In this table, we also include the oceanographic zones to which each of these diatom species are affiliated. This new table includes all the necessary information for the reader to understand that among the identified diatoms, diatoms from the NAZ account for the largest proportion of the main diatom assemblage.

**References**

Allen, C. S., Thomas, E. R., Blagbrough, H., Tetzner, D. R., Warren, R. A., Ludlow, E. C., & Bracegirdle, T. J. (2020). Preliminary evidence for the role played by South Westerly wind strength on the marine diatom content of an Antarctic Peninsula Ice Core (1980–2010). Geosciences, 10(3), 87.

Tetzner, D. R., Thomas, E. R., & Allen, C. S. (2021). Marine diatoms in ice cores from the Antarctic Peninsula and Ellsworth Land, Antarctica–species diversity and regional variability. The Cryosphere Discussions, 1-32.

**A. Reviewer 2 comment #4**

Based on the spatial correlations, this new tool gives an idea of wind strength changes in very small regions of the SWW core. It however gives no information on important aspects of the SWW system, i.e. whether changes in strength are associated to changes in the intensity or the position/expansion of the SWW core, or in the winds' direction that may sweep different regions as shown by the rich diatom assemblages (many coastal and few open ocean diatoms). In conclusion, I wonder whether this is possible to really deconvoluate between wind strength, wind direction and source areas as potential drivers of diatom abundances in ice cores. Not to speak about variable diatom production in different oceanic realms, potential depletion of benthic diatoms from wet rocks, ice, etc….

This comment includes some points that have been previously addressed. Comments regarding the position and size of the area of spatial correlation have been previously addressed in response to reviewers comment #1, section #1.1 and section #1.2. Comments regarding the diatom assemblage have been previously addressed in response to reviewers comment #3.

The reviewer states that the regionally consistent area of spatial correlation between diatom abundance in ice cores and wind strength does not give information about the SHWW system. We disagree with the reviewer's comment. Our results show the diatom abundance in high elevation ice core sites exhibit a high, statistically significant, correlation with wind strength over a large area within the Pacific core of the SHWW (55-60°S, 120-150°W). This relationship establishes a direct link between SHWW intensity and the number of diatoms preserved in ice core layers from high elevation AP sites. Similarly, the main diatom assemblage demonstrates to hold valuable information about the principal diatom source. The current diatom source lies over the NAZ, contributing with a characteristic diatom assemblage to ice core sites. If the core of the SHWW was positioned slightly north (50-55°S) or south (60-65°S), the assemblage would be considerably different. Thus, our results suggest the diatom record preserved in these ice cores hold unique information to track changes in SHWW intensity (through the diatom abundance) and in SHWW migration (through the main diatom assemblage).

Our results are based on the analyses of the wind speed parameter (not wind direction), mainly because the wind speed is the primary factor driving the transfer of diatoms from the oceanic surface to the atmosphere, through sea-spray production processes. These processes occur regardless of the wind direction. However, since the main source of diatoms is located within the core of the SHWW and the strong westerly winds from the core are the most likely to produce sea-sprays, it can be assumed that the wind direction at the source will remain stable in time (westerlies).

Based on the location of the source (NAZ), it can be assumed that there is no considerable variability in the diatom production (**Tetzner et al., 2021**).

**References**

Tetzner, D. R., Thomas, E. R., & Allen, C. S. (2021). Marine diatoms in ice cores from the Antarctic Peninsula and Ellsworth Land, Antarctica–species diversity and regional variability. The Cryosphere Discussions, 1-32.

**A. Reviewers comment #5**

Overall, I am very puzzled about the QHCs localisations in the middle of the Pacific sector of the Southern Ocean, which does not fit with the back-trajectories (Allen et al., 2020) and diatom assemblages preserved in the ice cores (Tetzner et al., 2021a). Some elaborations on these aspects would be welcome.

**Response:**

The QHCs from high elevation ice cores presented in this manuscript do in fact fit within the back-trajectory region presented by **Allen et al. (2020)** and also coincide with the area of high correlation identified by **Allen et al. (2020)** for the neighbouring Ferrigno ice core (See response to reviewers comment #1, section #1.2). Likewise, the diatom ecological affiliations obtained for JUR and SKBL support an open ocean source, south (and/or within) the Antarctic Polar Front, as the primary source of diatoms to the ice core sites. Thus, coincident with the oceanographic conditions that prevail in the surroundings of the QHCs (See response to reviewers comment #3).

To address this comment, we have modified Figure 3 to show the wider area of statistically significant correlations (See response to comment #1.1) (**See new Figure 3**). We have also included a table in Appendix A (**Table A1**) to let the reader know which diatoms shape the main diatom assemblage at each ice core site and the aquatic environments that they inhabit.

**References**

Allen, C. S., Thomas, E. R., Blagbrough, H., Tetzner, D. R., Warren, R. A., Ludlow, E. C., & Bracegirdle, T. J. (2020). Preliminary evidence for the role played by South Westerly wind strength on the marine diatom content of an Antarctic Peninsula Ice Core (1980–2010). Geosciences, 10(3), 87.

**A. Reviewers comment #6**

Half of the references in the Introduction are auto-citations. For example, there are many other studies showing the recent warming in AAP based on instrumental data (lines 24-25). Similarly, there are other groups working with climate reanalyses.

**Response:**

We agree with the reviewers comment. To address this comment, we incorporated additional references in the Introduction section to highlight the valuable work that other research groups have done in this region (**Lines 24, 35, 38**).

**A. Reviewers comment #7**

In Allen et al. (2020), fragments of large diatoms were included in the total diatom content. I could not find this information in Tetzner et al. (2021a) or in the present study. As such, I am unable to evaluate whether the total diatom abundance is robust or not, as one large diatom can form several fragments. And it is impossible to evaluate whether such fragmentation occurs in surface water, during depletion and transport or during precipitation at the ice core site.

**Response:**

The diatom abundance parameter presented in this manuscript and **Tetzner et al. (2021)** include all diatom frustules and indistinctive diatom fragments found on each ice core, in line with the results presented by **Allen et al. (2020)**. This information has been recently incorporated in **Tetzner et al. (2021)** (https://doi.org/10.5194/tc-2021-160-AC1).

Diatom fragments are found on each ice core site presented in this work and are commonly reported in diatom records preserved in Antarctic ice cores (**Burckle et al., 1988a; Burckle et al., 1988b; Budgeon et al., 2012; Delmonte et al., 2017**). The recovery of specimens still articulated in short chains at the bottom of the ice cores presented in this work (**Tetzner et al., 2021**), evidence the diatoms we find in ice cores are not affected by mechanical fracturing after being deposited. Thus, evidencing fragmentation must occur before diatoms reach the ice core sites, either while transported in the atmosphere, suspended in aquatic environments or after being exposed to sub-aerial environments. This information has been recently incorporated in **Tetzner et al. (2021)** (https://doi.org/10.5194/tc-2021-160-AC1).

To address this comment, we have modified the manuscript (**Lines 125 and 130**) to specify that:

"Diatom counts per sample (n) included all valves, partially obscured valves and diatom fragments identified in each sample"

and

"The diatom abundance parameter includes all diatoms and diatom remains identified on each sample, regardless of their potential source"

**References**

Allen, C. S., Thomas, E. R., Blagbrough, H., Tetzner, D. R., Warren, R. A., Ludlow, E. C., & Bracegirdle, T. J. (2020). Preliminary evidence for the role played by South Westerly wind strength on the marine diatom content of an Antarctic Peninsula Ice Core (1980–2010). Geosciences, 10(3), 87.

Burckle, L. H., Gayley, R. I., Ram, M., & Petit, J. R. (1988a). Diatoms in Antarctic ice cores: some implications for the glacial history of Antarctica. Geology, 16(4), 326-329.

Burckle, L. H., Gayley, R. I., Ram, M., & Petit, J. R. (1988b). Biogenic particles in Antarctic ice cores and the source of Antarctic dust. Antarctic Journal of the United States, 23(5), 71-72.

Budgeon, A. L., Roberts, D., Gasparon, M., & Adams, N. (2012). Direct evidence of aeolian deposition of marine diatoms to an ice sheet. Antarctic Science, 24(5), 527-535.

Delmonte, B., Paleari, C. I., Andò, S., Garzanti, E., Andersson, P. S., Petit, J. R., ... & Maggi, V. (2017). Causes of dust size variability in central East Antarctica (Dome B): Atmospheric transport from expanded South American sources during Marine Isotope Stage 2. Quaternary Science Reviews, 168, 55-68.

Tetzner, D. R., Thomas, E. R., & Allen, C. S. (2021). Marine diatoms in ice cores from the Antarctic Peninsula and Ellsworth Land, Antarctica–species diversity and regional variability. The Cryosphere Discussions, 1-32.

---

## Author Response (AR2)

Dear Editor,

We thank reviewers #1 and #2 for their time and consideration in reviewing our manuscript. We are pleased that anonymous reviewer #1 recognises the value of this work and recommends our manuscript should be published. Please find to follow our response to anonymous reviewer #2 comments. Reference lines mentioned in our response correspond to lines in our modified manuscript (version with "track changes" disabled).
* * *
**Reviewer #2 comment 1:**

"I am however still confused by the diatom assemblages preserved in the ice cores and how they relate to the ocean environments in comparison to the spatial correlation maps. In SHIC core, >60% of the diatoms are made by *Fragilariopsis cylindrus* (Tetzner et al., 2021). This species is linked to heavy sea ice conditions both in plankton and surface sediments, i.e. rather towards continental shelf and bay systems (Kang and Fryxell, 1992 ; Leventer et al., 1993 ; Burckle et al., 1987 ; Armand et al., 2005 ; Beans et al., 2008 ; Esper et al., 2010 ; Campagne et al., 2016). Additionally, *F.cylindrus* is very abundant in the Amundsen and Bellinghausen coastal regions due to the presence of coastal polynyas (Kellog and Kellog, 1987). So, if *F. cyclindrus* is responsible of the correlation with sea ice cover (fig 3), I wonder how/why the QHC is towards the winter sea ice edge where *F. cyclindrus* is probably not very abundant. I also wonder how *F. cyclindrus* can be wind blow from a region where it is not present at all, as the wind QHC is far north of the winter sea ice limit.

I agree that there is a statistically significant correlation between diatom abundances and winds, but the ecological preferences of the main diatom species transported to SHIC ice core would argue against a long distance transport. I wonder whether *F. cylindrus* (and other coastal diatoms) could be entrained by strong winds in the coastal region. The link between these winds and the SHWW needs to be discussed."

**Response to comment 1:**

We agree with Reviewer #2 that the diatom assemblage of SHIC is not derived by long-distant transport but is sourced more locally, from within the seasonal sea ice zone (SSIZ). A proximal SSIZ source of diatoms for SHIC is supported by the high diatom concentration and strong seasonal variability, reflecting the intense seasonal blooms that characterize the SSIZ (Tetzner et al., 2022). Additionally, the proximity of the summer sea ice edge (SIE) and coastal polynyas to the SHIC, provide the most likely source for the diatoms and account for the dominance of F. cylindrus.

Based on this evidence, we explicitly state that the coastal areas exhibiting high correlations along the Bryan coast and the Amundsen Sea coast (Figure 1 and Figure 3) are the most likely source regions for diatoms in the SHIC (as suggested in reviewer's #2 comment) (Manuscript lines 276-315).

To address this comment, we have amended the abstract and text throughout (Lines 283-315) in order to reiterate the proximal source of diatoms to the SHIC:

Example (Lines 292-295): "*Previous studies have identified recent reductions in the ice-covered days (sea ice concentration) during the austral summer (Stammerjohn et al., 2012) and the development of*

*coastal polynyas (Eltanin Polynya, Pine Island Polynya and Amundsen Sea Polynya) within the __coastal__ regions identified as the SHIC diatom sources (Arrigo and van Dijken, 2003; Arrigo et al., 2012)*".

We have also added text to better explain the area(s) of high correlation in the northern Bellingshausen and Amundsen Seas (Lines 278-281), which likely reflect(s) the inter-connectedness of regional atmospheric circulation (ie. zonal and meridional winds, and sea ice cover/distribution), over the Amundsen & Bellingshausen Seas.
* * *
**Reviewer #2 comment 2:**

"Similarly, coastal diatoms (*Cyclotella* gp, *Achnanthes* gp and *Navicula* gp) amount 40% and >50% of the total diatom assemblages preserved in SKBL and JUR, respectively. I do not understand how they can be entrained from the open ocean where the QHC lies. I believe they are rather wind blown from the coastal regions (Amundsen/Bellinghausen or South America). Their link to the QHC in the core SHWW is therefore not clear at all and should better discussed. For example, I woudl suggest to assess whether the meriodional component of the Admunsea Low (potential regional entrainment force) is always in phase with the SHWW speed changes because, again, SHWW can't transport these coastal diatoms to the ice cores. Another test/approach would be to assess the relationship between the sole open ocean diatoms with wind speed"

**Response to comment 2:**

The main diatom assemblage of JUR and SKBL presents considerable amounts of *Cyclotella* gp, *Achnanthes* gp and *Navicula* gp. These diatom genera have a broad distribution and can be found across freshwater, brackish, and marine environments. Our results highlight a region of high correlation between the diatom abundance at JUR and SKBL and wind speed at the northern edge of the Amundsen Sea and within the core of the Southern Hemisphere Westerly Wind (SHWW) belt (~60S). Regional air mass back trajectory analyses show airmasses reaching high elevation ice core sites on the peninsula are at sea level only a considerable distance over the ocean and are higher in the atmosphere when they traverse the Antarctic Peninsula coast (Thomas and Bracegirdle, 2015; Allen et al., 2020). These results suggest that diatoms transported to JUR and SKBL are either removed from the sea surface within the SHWW belt or from lower latitudes and then entrained by air masses moving south. The back trajectory results show it is unlikely that air masses actively entraining particles from coastal sites in the Antarctic Peninsula would transport them to inland ice core sites in the Antarctic Peninsula (JUR and SKBL). Although we cannot totally rule out a local source for the *Cyclotella*, *Achnanthes* and *Navicula* groups recovered from JUR and SKBL, the available evidence suggests these diatoms are likely entrained in the SHWW and/or lower latitudes.

To address this comment we have modified the manuscript to emphasize that air masses reaching inland ice core sites only interact with the ocean surface away from the coast (Lines 324-326).